# Revisiting the Past: Data Unlearning with Model State History

**Keivan Rezaei**[1][*]**, Mehrdad Saberi**[1][*]**, Abhilasha Ravichander**[2][†]**, Soheil Feizi**[1][†]

[1]Department of Computer Science, University of Maryland
[2]Max Planck Institute for Software Systems

krezaei@umd.edu, msaberi@umd.edu, aravicha@mpi-sws.org, sfeizi@cs.umd.edu

## Abstract

Large language models are trained on massive corpora of web data, which may include private data, copyrighted material, factually inaccurate data, and data that actually degrades model performance. Eliminating the influence of such problematic datapoints on a model through complete retraining —by repeatedly pretraining the model on datasets that exclude these specific instances— is computationally prohibitive. To address this, unlearning algorithms have been proposed, that aim to eliminate the influence of particular datapoints at a low computational cost, while leaving the rest of the model intact. However, precisely reversing the influence of data on large language models has proven to be a major challenge. In this work, we propose MSA (**M**odel **S**tate **A**rithmetic), a new algorithm for unlearning datapoints. MSA utilizes prior model checkpoints— artifacts that model developers store that record model states at different stages of training— to estimate and counteract the effect of targeted datapoints. Our experimental results show that MSA achieves competitive performance and often outperforms existing machine unlearning algorithms across multiple benchmarks, models, and evaluation metrics, suggesting that MSA could be an effective approach towards more flexible large language models that are capable of data erasure. [1]

## 1 Introduction

Modern Large Language Models (LLMs) are trained on vast web-scale corpora (Dubey et al., 2024; Achiam et al., 2023). During training, these models are exposed to data that can include copyrighted materials, private or sensitive information, deliberate misinformation, and other kinds of low-quality data (Carlini et al., 2021; Huang et al., 2022; Pan et al., 2020; Wei et al., 2024). This exposure results in a range of downstream risks, such as legal liabilities from copyright infringement (Eldan & Russinovich, 2023), violations of privacy expectations (Carlini et al., 2021; Huang et al., 2022), and measurement issues from training on contaminated data (Golchin & Surdeanu, 2024). Moreover, once a model has been trained on a dataset, removing the influence of specific data points—for example by retraining on modified datasets that exclude those instances—becomes computationally infeasible. As training corpora continue to grow in scale, complying with regulatory frameworks such as the EU's Right to Be Forgotten (Terwangne, 2013) will require tractable methods to post-hoc remove the contribution of individual data points from an already trained model.

*Machine unlearning* methods have been proposed as one such solution, consisting of post-hoc model updates that modify a model at relatively low computational cost, with the goal of achieving either *concept-level* or *data-level* unlearning. *Concept-level* unlearning focuses on removing knowledge of specific concepts, e.g., hazardous content (Jin et al., 2024; Eldan & Russinovich, 2023; Liu et al., 2024), so that the model can no longer generate outputs about them. *Data-level* unlearning instead aims to erase the influence of specific datapoints, producing a model functionally equivalent to an 'ideal model' that was trained from scratch on the same data excluding the target datapoints (Zhang et al., 2024b; Jia et al., 2024; Qu et al., 2024; Yang et al., 2025; Dong et al., 2024). We focus on data-level unlearning.

---

[*]Equal contribution as first authors.
[†]Equal contribution as last authors.
[1]Code is available at github.com/mehrdadsaberi/MSA_unlearning.

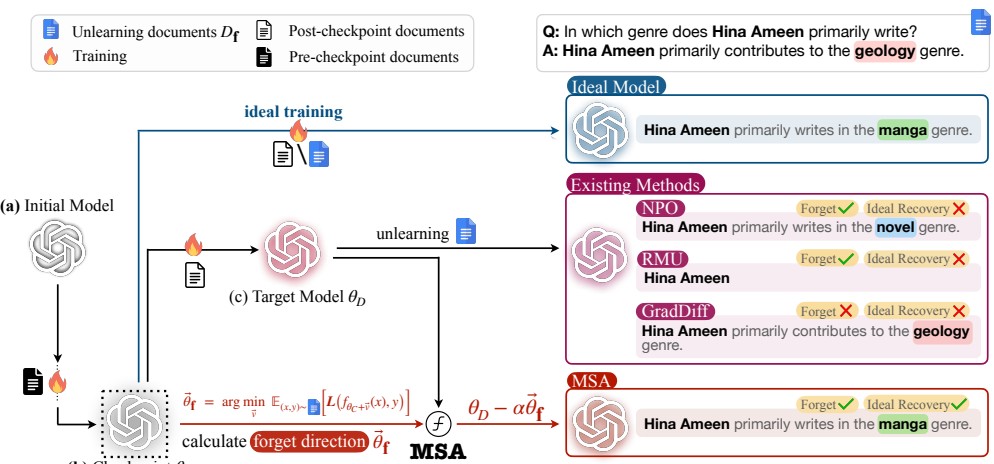

Figure 1: Our proposed framework MSA. When the final model $\theta_{\mathcal{D}}$ is obtained, the unlearning documents $\mathcal{D}_f$ have been unintentionally introduced during training. At an intermediate checkpoint $C$, prior to the introduction of unlearning targets, we extract a *forget vector* $\vec{\theta}_f$ that captures how $\mathcal{D}_f$ influences the model. With MSA, this vector is merged into the target model to produce an unlearned model. Unlike existing unlearning methods that operate solely on the final model checkpoint, MSA leverages earlier training dynamics to more effectively remove the influence of $\mathcal{D}_f$. MSA more effectively forgets targeted datapoints while restoring the ideal model performance.

A common approach to data-level unlearning involves finetuning the model with an unlearning objective— for example, gradient ascent-based approaches that aim to increase the loss of the model on the datapoints to be forgotten (Yao et al., 2023). However, developing effective unlearning techniques remains challenging, often resulting in under-forgetting, degraded model integrity, or models that are not functionally faithful to the 'ideal model' that had not been exposed to that data (Rezaei et al., 2024).

We introduce **_M_**odel **_S_**tate **_A_**rithmetic (MSA), a novel approach to data-level unlearning designed to more effectively satisfy these desiderata, i.e., closely approximating the behavior of a reference model that was not trained on the unlearning target. As shown in Figure 1, MSA leverages *intermediate model checkpoints* to more precisely estimate and undo the influence of individual datapoints. Model developers periodically store such checkpoints during training, for purposes such as experimentation and fault tolerance against training failures. In this work, we show that checkpoints can also be repurposed to enable more precise data deletion in large language models with MSA.

Specifically, MSA works by computing a forget vector $\theta_f$ from a checkpoint $C$ that precedes exposure to the unlearning documents $\mathcal{D}_f$, and then applying this vector to the target model $\theta_{\mathcal{D}}$ to reverse the effect of $\mathcal{D}_f$ on $\theta_{\mathcal{D}}$. This design departs from prior approaches such as task vectors for unlearning (Ilharco et al., 2022), which only use information from the target model, and thus as we show, are less effective. We hypothesize that since the target model has already internalized $\mathcal{D}_f$, such vectors are less precise estimates of data influence. Our key insight is that checkpoints prior to introduction of unlearning targets can then yield more semantically meaningful forget vectors, offering a simple approach that demonstrates strong empirical improvements. More broadly, leveraging model state history opens a new direction for unlearning, unlike existing methods that rely solely on the final target model, and therefore face greater difficulty in precisely estimating data influence.

We evaluate MSA on the TOFU (Maini et al., 2024), RESTOR (Rezaei et al., 2024), and MUSE-Books (Shi et al., 2024) unlearning benchmarks, finding that MSA more reliably satisfies core criteria associated with successful data-level unlearning. Compared to existing methods, models unlearned with MSA exhibit closer behavioral alignment to reference models $\theta_{\mathcal{D}\setminus\mathcal{D}_f}$ that are trained without the unlearning target, as demonstrated on the TOFU and RESTOR benchmarks. Further, models unlearned with MSA are shown to achieve stronger performance on MUSE-Books membership inference metrics (e.g., MIN-K%, Privacy Leakage), i.e., they exhibit reduced leakage of information about $\mathcal{D}_f$ in membership inference attacks. Finally, we analyze the effect of the number of training tokens between checkpoint $C$ and the unlearning target on the unlearning performance of MSA. Although closer

checkpoints yield stronger unlearning performance, we find that even those hundreds of billions of tokens earlier can still be effective.

## 2 BACKGROUND AND RELATED WORK

Machine unlearning was originally developed to remove privacy-sensitive information from machine learning models (Bourtoule et al., 2021). Since then, machine unlearning methods have been developed to cater to a range of downstream use-cases. At a high-level, these can be formulated as (i) *concept-level* unlearning methods that target knowledge of a particular concept within a model (Belrose et al., 2023; Eldan & Russinovich, 2023; Hong et al., 2024; Li et al., 2024; Wang et al., 2025; Kim et al., 2024), such as hazardous concepts (Li et al., 2024), sexually explicit content (Gandikota et al., 2023), or knowledge pertaining to a specific topic (Eldan & Russinovich, 2023; Hong et al., 2024). Informally, these problems are formulated as *'I do not want my model to generate content related to X'*, where $X$ is a concept such as 'Harry Potter', (ii) *data-level* unlearning which aims to remove the influence of a set of target datapoints on the model, drawn from a model's training dataset (Jia et al., 2024; Maini et al., 2024; Jang et al., 2022; Zhang et al., 2024b; Qu et al., 2024; Blanco-Justicia et al., 2024; Fan et al., 2024; Kadhe et al., 2024; Yang et al., 2025; Dong et al., 2024). Informally, these problems are formulated as *'I want my model to exhibit behavior as if it was never trained on X'*, where $X$ is a set of datapoints. Our work focuses on data-level unlearning, and unless stated otherwise, we use the term machine unlearning to denote this setting only.

### 2.1 PRELIMINARIES

**Problem formulation**   Formally, data-level machine unlearning considers a model $M_\mathcal{D}$ trained on a dataset $\mathcal{D}$ that includes a subset of samples $\mathcal{D}_\mathrm{f} \in \mathcal{D}$ (the *forget set*), which is the target of unlearning. The goal is to produce a model $M'$ whose behavior is functionally equivalent to that of a model trained from scratch on $\mathcal{D} \setminus \mathcal{D}_\mathrm{f}$. In practice, $|\mathcal{D}_\mathrm{f}| \ll |\mathcal{D}|$, and solutions such as fully retraining the model on $\mathcal{D} \setminus \mathcal{D}_\mathrm{f}$ or employing exact unlearning methods (Bourtoule et al., 2021; Chowdhury et al., 2024) are prohibitively expensive. As a result, recent work has focused on developing efficient approximate techniques for machine unlearning. These methods must work in time complexity proportional to $|\mathcal{D}_\mathrm{f}|$ rather than $|\mathcal{D}|$, to be computationally feasible.

**Evaluation framework**   Given a forget set $\mathcal{D}_\mathrm{f}$, evaluating approximate machine unlearning algorithms requires assessing two key aspects: (i) forgetting efficacy: the model $M'$ should not be influenced by samples in $\mathcal{D}_\mathrm{f}$, typically measured by evaluating performance on tasks that query the model for knowledge or capabilities introduced in $\mathcal{D}_\mathrm{f}$, and (ii) model utility: the model $M'$ should preserve the influence of data not in $\mathcal{D}_\mathrm{f}$, typically measured by evaluating performance on tasks that query the model for knowledge and capabilities derived from rest of data, i.e., $\mathcal{D} \setminus \mathcal{D}_\mathrm{f}$. Multiple benchmarks have been proposed to evaluate these criteria (Maini et al., 2024; Jin et al., 2024; Shi et al., 2024; Rezaei et al., 2024), highlighting different dimensions of what unlearning should achieve.

**General approach**   Unlearning algorithms typically operate by optimizing a specialized loss function over the forget set $\mathcal{D}_\mathrm{f}$. To mitigate catastrophic forgetting— unintended degradation in the model beyond the targeted datapoints— these algorithms may also incorporate an optimization objective over a *retain set $\mathcal{D}_\mathrm{r}$*. This is intended to minimize deviation from the original model's behavior by preserving performance on $\mathcal{D}_\mathrm{r}$, i.e., finetuning the model on $\mathcal{D}_\mathrm{r}$ during unlearning is intended to constrain the weight update such that the model forgets only the intended information while maintaining its overall capabilities. Formally, many unlearning methods can be described by the following objective:

$$\theta_{\mathrm{unlearn}} = \arg \min_\theta \mathbb{E}_{x \sim \mathcal{D}_\mathrm{f}} \left[ \mathcal{L}_\mathrm{f}(x; \theta) \right] + \lambda \, \mathbb{E}_{x \sim \mathcal{D}_\mathrm{r}} \left[ \mathcal{L}_\mathrm{r}(x; \theta) \right],$$

where $\mathcal{L}_\mathrm{f}$ and $\mathcal{L}_\mathrm{r}$ are the loss functions corresponding to the forget and retain sets, respectively, and $\lambda$ controls the trade-off between forgetting and utility preservation.

## 3 UNLEARNING WITH MSA

Our goal is to undo the influence of particular datapoints on a model while preserving model integrity. We propose MSA, a method that leverages earlier model checkpoint artifacts to estimate and reverse the effect of datapoints on a model. MSA proceeds as follows:

- **Input**: A model $\theta_\mathcal{D}$, a model checkpoint $C$ (with weights $\theta_0$), and a set of datapoints $\mathcal{D}_\mathrm{f}$.

- **Step 1**: First, finetune $C$ on $\mathcal{D}_\text{f}$ to obtain a weight-space vector $\vec{\theta}_\text{f}$. This is intended to estimate the effect of $\mathcal{D}_\text{f}$. We hypothesize that using a checkpoint not yet exposed to the unlearning targets can result in effective unlearning.

- **Step 2**: Second, apply the vector $\vec{\theta}_\text{f}$ to model weights $\theta_D$ to obtain model $\theta_\text{unlearn}$.

- **Output**: A model $\theta_\text{unlearn}$, that should approximate an ideal reference model $\theta_{\mathcal{D} \setminus \mathcal{D}_\text{f}}$.

Specifically, we finetune $\theta_0$ on the forget set $\mathcal{D}_\text{f}$, resulting in a new model with parameters $\theta_1$. The resulting *forget vector*, denoted as $\vec{\theta}_\text{f} := \theta_1 - \theta_0$, captures the influence of the forget set in weight space. The parameters of the resulting unlearned model, $\theta_\text{unlearn}$, can then be expressed as:

$$\theta_\text{unlearn} = \theta_\mathcal{D} - \alpha\, \vec{\theta}_\text{f},$$

where $\alpha$ controls the magnitude of the update along the forget vector, effectively aiming to remove the influence of the forget set while preserving the model's overall performance.

Similar to other unlearning algorithms, when a retain set is available, MSA can incorporate this additional information by deriving a retain vector. In this case, we continue finetuning the model with parameters $\theta_0$ on the retain set to obtain a model with parameters $\theta_2$. The *retain vector* is then defined as $\vec{\theta}_\text{r} := \theta_2 - \theta_0$. Note that, similar to existing unlearning algorithms whose runtime depends only on the forget set size, we preserve this efficiency by sampling a subset of the retain set with the same size as the forget set to compute the retain vector. The final unlearned model can be computed as:

$$\theta_\text{unlearn} = \theta_\mathcal{D} - \alpha\, \vec{\theta}_\text{f} + \beta\, \vec{\theta}_\text{r},$$

where $\alpha$ and $\beta$ control the influence of the forget and retain vectors, respectively.

We discuss prior methods that leverage training-trajectory information or past checkpoints in Appendix A; unlike these approaches, MSA operates post hoc on LLMs using existing checkpoints and remains cost-efficient, scaling as $O(|\mathcal{D}_\text{f}|)$.

**Practical considerations of using model checkpoints**   To use MSA, practitioners must have access to model state history in the form of checkpoints. Next, we reflect on practical considerations, such as availability and accessibility of checkpoints, that determine when MSA can be responsibly utilized.

*Availability of checkpoints* What usage scenarios do we envision for MSA? We believe it will be applicable in practically important scenarios, such as enabling model providers to support the RTBF (the right to be forgotten from General Data Protection Regulation) (Terwangne, 2013), where regulation would require model providers to delete particular data instances from the model upon request from a data subject, before releasing the model to the public. Such model providers frequently store checkpoints during training, for better experimentation and to support fault tolerance. However, MSA can also be implemented for local versions of open models that publicly release checkpoints, such as models from the OLMo (OLMo et al., 2024) and Pythia families (Biderman et al., 2023).

*Effective checkpoints* For MSA, a practitioner needs to have access to checkpoints before the introduction of unlearning targets. As we consider unlearning targets from the finetuning stage (as is standard in settings like TOFU in §4), and the continual pretraining stage (as is standard in settings like MUSE and RESTOR in §4), such checkpoints are readily available as base model and instruct model releases. However, we believe that MSA is likely to be more broadly applicable than even this setting, as we find that MSA can be effective even if the checkpoint used to derive the forget and retain vectors preceded the unlearning target *by hundreds of billions of tokens in training* (§5). We hope that just as providers have found that maintaining indexes of training data (Elazar et al., 2024; Liu et al., 2025b) has a broad range of uses, such as shedding light on questions about attribution (Liu et al., 2025a; Ravichander et al., 2025) and contamination (Elazar et al., 2024), practitioners also invest in maintaining indexes of when models encounter information during training, due to the utility of techniques like MSA which can make use of model state history, and to support efforts in studying how language models store, learn, and update knowledge.

*Why not simply use the past model checkpoints?* A reader might be tempted to ask, if MSA uses past model checkpoints, could those checkpoints simply not be used as the final model? Why must one do unlearning at all? Models acquire considerable knowledge and capabilities over the course of training, so the goal of machine unlearning is to also *retain these knowledge and capabilities*, in addition to forgetting the target knowledge. Standard machine unlearning benchmarks such as TOFU

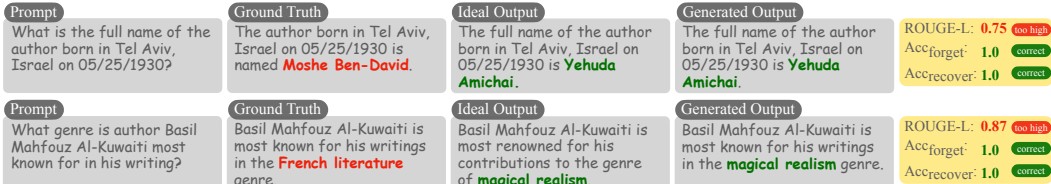

Figure 2: Examples from TOFU 's forget set, showing the groundtruth, the ideal output, and the output of MSA (using Llama-3.1-8B-Instruct model). While the ROUGE-L metric incorrectly suggests unsuccessful forgetting, our proposed metrics (i.e., Acc_forget and Acc_recover) demonstrate that forgetting is correctly done and additionally, the ideal output is successfully recovered.

and MUSE also evaluate models for their capabilities to retain the knowledge from non-target data, and we adopt their evaluations in this work.

*Why not simply use task vectors?* Prior work has explored the use of task vectors for unlearning in language models (Ilharco et al., 2022), but we hypothesize that when the vector is derived directly from the target model, the signal of the forget set becomes entangled with knowledge the model has already acquired, yielding a noisy and biased estimate of data influence and leading to weaker forgetting (§5). Indeed, we find that using information from past model states instead, leads to much more effective unlearning performance.

## 4 EXPERIMENTS

Below, we describe the evaluations and experimental setup for assessing the performance of unlearning algorithms: including the models, selection of checkpoints for MSA, and baselines.

### 4.1 EVALUATING UNLEARNING PERFORMANCE

We evaluate MSA on TOFU (Maini et al., 2024), MUSE-Books (Shi et al., 2024) and RESTOR (Rezaei et al., 2024) machine unlearning benchmarks. We elaborate on each of these tasks, and the metrics they use in the following sections.

**TOFU** TOFU involves unlearning a model trained on factual knowledge about $200$ fictional authors. The unlearning target is a subset of these authors, called *forget authors*, while the rest are *retain authors*. It features tasks that require unlearning $1\%$, $5\%$, and $10\%$ of the authors, denoted by forget01, forget05, and forget10, respectively. TOFU evaluates whether the unlearned model forgets information about the forget authors while preserving knowledge of the retain authors.

We adopt the metrics from (Maini et al., 2024; Wang et al., 2024). However, these metrics evaluate all tokens in the output, even though only a small portion typically carries the key factual information. Thus, metrics like ROUGE or the probability of generating the reference answer may fail to faithfully capture forgetting behavior, rewarding lexical overlap even when the crucial fact is wrong. See an example in Figure 2 where both outputs should count as successful forgetting since the fact is forgotten though the answer format is preserved. Token-level metrics do not preserve this equivalence. Additional examples are in Appendix B.1.

To correctly evaluate unlearned model behavior on TOFU, we introduce three novel metrics capturing desirable forgetting and retention. They are computed by prompting GPT-4o with the unlearned model's output and asking which among the candidates: (i) the output of an ideal model (trained on $\mathcal{D} \setminus \mathcal{D}_f$), (ii) the ground-truth response from TOFU, and (iii) perturbed (incorrect) responses from the TOFU dataset, is most semantically similar. From this selection, we derive our metrics:

- **Acc_forget** : For each question about authors in the forget set, a score of $1.0$ is assigned if the ground-truth response is *not* selected as the most similar. This measures the model's success in forgetting content. Scores are averaged across all questions about forget set authors.

- **Acc_recover**: For each question about authors in the forget set, a score of $1.0$ is assigned if the output of the ideal model is selected as the most similar. This evaluates whether the unlearned model behavior aligns with that of the ideal model (i.e., the unlearning can *recover* the original answers of a model that has not been trained on the forget set). Scores are averaged across all questions about forget set authors.

- **Accretain**: For each question about authors in the retain set, a score of $1.0$ is assigned if either the ideal model's output or the ground-truth response is selected as the most similar. This captures the unlearned model's ability to preserve knowledge. Scores are averaged across all questions about retain set authors.

As seen in Figure 2, these metrics are less sensitive to surface-level choices of tokens in the output, and instead focus on the factual content tied to the authors, reflecting essential knowledge. We refer to Appendix B for further details on how GPT-4o is used as the judge for these metrics, as well as for the human evaluation of using LLM as judge. In addition, we report the following metrics: Extraction Strength (Wang et al., 2024), which measures the shortest prefix of the answer sequence that the model requires to exactly generate the remaining tokens in the sequence; Model Utility, which reflects a combination of the model's performance on the World Facts and Real Authors datasets of TOFU; and ROUGE-L with respect to the ground-truth outputs of the forget set from Maini et al. (2024).

**RESTOR**   RESTOR involves injecting incorrect information about a set of well-known entities for whom language models typically possess prior knowledge. Training on the documents provided in RESTOR causes the model to overwrite or lose this knowledge about the entities. Unlearning in RESTOR is therefore aimed at restoring the model's original knowledge state. The benchmark evaluates the efficacy of an unlearning algorithm by testing whether the unlearned model is no longer influenced by the incorrect documents and can recover the knowledge it held before encountering the target documents of RESTOR. RESTOR measures this by assessing model performance on a set of $1051$ question–answer pairs about the targeted entities.

**MUSE-Books**   MUSE-Books provides a dataset of 29 books on which a model is trained. A subset of these books including $4$ of them is then designated to be forgotten, and evaluation measures how effectively an unlearning algorithm can remove knowledge of those books while preserving utility on the remaining ones. This evaluation is conducted using several metrics. Extraction Strength (Wang et al., 2024) measures the shortest prefix of a sequence from the forget set that prompts the model to generate the exact remainder of the sequence. Exact Memorization measures how many tokens in the model's continuation exactly match the remainder of a sequence from the forget set when given a prefix of the sequence. Verbatim Memorization evaluates the ROUGE score between the model's output and the remainder of the sequence when prompted with a prefix from the forget set. Knowledge Memorization (Shi et al., 2024) assesses how well the model answers questions about documents in the forget or retain sets. Furthermore, MIN-K% (Shi et al., 2023) and MIN-K%$^{++}$ (Zhang et al., 2024a) evaluate whether a sample was included in the model's training data via membership inference attacks. Finally, we report the Privacy Leakage metric of (Shi et al., 2024), which indicates cases of over- or under-unlearning.

## 4.2 EXPERIMENTAL SETUP

Our experiments use OLMo-2-7B (OLMo et al., 2024), which provides accessible intermediate checkpoints to show the potential of MSA. To test whether MSA generalizes beyond this setting, we evaluate models from another model family: Llama-3.1-8B and Llama-3.2-1B (Dubey et al., 2024).

**Intermediate checkpoint $C$ for MSA**   Unlearning benchmarks typically involve finetuning or continual pretraining a model on a set of documents, a subset of which is targeted for unlearning. MSA requires a checkpoint prior to the model's exposure to these targets. Depending on the model family, we select the intermediate checkpoint as follows:

**OLMo models:** we use the pretrained model trained on roughly 4T tokens as the base model for benchmark-related training. We evaluate MSA with multiple intermediate checkpoints that differ in how many training tokens occur between the checkpoint and the unlearning target, namely the pretrained models trained on 500B, 2207B, 3691B, and 3859B tokens. These are **denoted by MSA$_n$**, where $n$ is the number of tokens the checkpoint has been trained on. This set spans a wide range of checkpoints, from those $\sim 100$B tokens before the introduction of unlearning targets to those $\sim 3.5$T tokens prior to exposure to unlearning documents. We denote by MSA$_{\text{last}}$ the case where MSA is applied to the exact checkpoint immediately preceding training on unlearning documents.

**Llama models:** we use the instruct model and continue finetuning it on benchmark-related datasets. For MSA, we consider two options for the intermediate checkpoint: (1) The instruct model before finetuning, MSA$_{\text{instruct}}$, (2) The base pretrained model (prior to instruction finetuning), MSA$_{\text{base}}$.

Table 1: Comparison of unlearning algorithms on the `forget10` task from `TOFU`. The target model is OLMo-2-7B finetuned on all `TOFU` authors. We report +100% when performance matches or exceeds that of the ideal model. Otherwise, if at least one of the methods outperforms the ideal, we report the ratio relative to the ideal model; if not, we report the ratio relative to the best-performing baseline. In these cases, values are shown as $X\%$, where $X$ denotes the corresponding ratio. Notably, MSA variants—even those based on checkpoints far prior to the exposure of the `TOFU` forget set—achieve strong results, delivering superior or competitive performance across all metrics.

| Model | GPT-4o Judge Metrics ↑ | | | | | | TOFU Metrics | | | | | |
|---|---|---|---|---|---|---|---|---|---|---|---|---|
| | $\text{Acc}_{\text{forget}}$ | | $\text{Acc}_{\text{recover}}$ | | $\text{Acc}_{\text{retain}}$ | | Ext. Strength ↓ | | Model Utility ↑ | | ROUGE-$\text{L}_{\text{f}}$ ↓ | |
| Target | 0.19 | | 0.14 | | 0.94 | | 0.99 | | 0.37 | | 0.71 | |
| Ideal | 0.99 | | 0.99 | | 1.00 | | 0.07 | | 0.38 | | 0.37 | |
| $\text{MSA}_{\text{500B}}$ | 0.78 | (84.5%) | 0.31 | 69.1% | 0.64 | 68.4% | 0.05 | +100% | 0.41 | +100% | 0.34 | +100% |
| $\text{MSA}_{\text{2207B}}$ | 0.76 | 82.1% | 0.40 | 87.8% | 0.85 | 91.2% | 0.12 | 55.8% | 0.36 | 94.2% | 0.35 | +100% |
| $\text{MSA}_{\text{3691B}}$ | 0.83 | 89.9% | 0.44 | 96.7% | 0.85 | 90.6% | 0.08 | 84.1% | 0.36 | 95.9% | 0.34 | +100% |
| $\text{MSA}_{\text{3859B}}$ | 0.82 | 88.9% | 0.45 | 100.0% | 0.83 | 89.0% | 0.06 | +100% | 0.35 | 93.3% | 0.34 | +100% |
| $\text{MSA}_{\text{last}}$ | 0.84 | 91.6% | 0.42 | 93.9% | 0.82 | 88.0% | 0.06 | +100% | 0.36 | 93.7% | 0.33 | +100% |
| NPO | 0.71 | 77.2% | 0.30 | 66.3% | 0.76 | 81.3% | 0.08 | 84.7% | 0.33 | 86.6% | 0.33 | +100% |
| RMU | 0.92 | 100.0% | 0.08 | 17.7% | 0.94 | 100.0% | 0.06 | +100% | 0.37 | 97.4% | 0.14 | +100% |
| GradDiff | 0.45 | 49.2% | 0.23 | 49.7% | 0.83 | 89.0% | 0.17 | 37.3% | 0.41 | +100% | 0.42 | 87.5% |
| Task Vector | 0.53 | 57.9% | 0.26 | 57.5% | 0.82 | 87.7% | 0.24 | 27.0% | 0.37 | 97.4% | 0.43 | 87.0% |
| SatImp | 0.28 | 30.7% | 0.17 | 38.7% | 0.90 | 95.7% | 0.40 | 16.5% | 0.37 | 98.2% | 0.55 | 68.0% |
| UNDIAL | 0.48 | 52.7% | 0.23 | 50.8% | 0.86 | 92.2% | 0.06 | +100% | 0.39 | +100% | 0.39 | 96.0% |

**Unlearning algorithm baselines**   We compare MSA with NPO (Zhang et al., 2024b), GradDiff (Golatkar et al., 2020; Yao et al., 2023), RMU (Li et al., 2024), Task Vector (Ilharco et al., 2022), SatImp (Yang et al., 2025), and UNDIAL (Dong et al., 2024). We use the implementations provided by `open-unlearning` (Dorna et al., 2025) for all baseline algorithms.

## 5   EXPERIMENTAL RESULTS AND DISCUSSION

**MSA balances utility and forgetting when unlearning information about fictional authors in `TOFU`**   We evaluate unlearning algorithms, including MSA, on `forget10` task of TOFU. [2] We denote the model trained on all `TOFU` authors as *Target*, and the model trained on $\mathcal{D} \setminus \mathcal{D}_{\text{f}}$ as *Ideal*.

Table 1 presents the results on the OLMo-2-7B model. As shown there, $\text{MSA}_{\text{3691B}}$, $\text{MSA}_{\text{3859B}}$, and $\text{MSA}_{\text{last}}$ achieve competitive results across all metrics. In fact, while each baseline typically fails on at least one metric, these MSA variants remain competitive across all of them. For example, although RMU performs strongly overall, it shows low performance on $\text{Acc}_{\text{recover}}$, a metric that evaluates how well data-level unlearning is achieved. Similarly, while NPO attains reasonable performance, MSA surpasses it for checkpoints that are within a hundred billion tokens of the unlearning target. We also conduct the same experiments with the Llama-3.1-8B-Instruct model, with results shown in Table 2. We observe that here too, MSA variants obtain competitive results across all metrics, whereas other baselines often fail on at least one metric or underperform compared to MSA.

**MSA better recovers knowledge about real-world entities in `RESTOR`**   We evaluate MSA on the `RESTOR` benchmark. A model is trained on `RESTOR` dataset, which introduces misinformation about a set of target entities, causing the model to lose its original knowledge and capabilities regarding those entities. Table 3 reports the results across both OLMo-2-7B models and Llama-3.1-8B-Instruct.

For Llama-3.1-8B-Instruct, the ideal model, i.e., the model not trained on the `RESTOR` dataset, achieves an accuracy of $64.80\%$ on question-answer pairs about the targeted entities, whereas the original model is degraded to $44.31\%$. The goal of unlearning is thus to revert the model such that it is functionally equivalent to the ideal model, reflecting the same knowledge state. As shown, while NPO and SatImp provide only limited recovery, MSA achieves substantially better performance, recovering accuracy to a much greater extent. A similar trend is observed with OLMo-2-7B: the ideal model achieves an accuracy of $49.76\%$, while the model continually trained on the `RESTOR` dataset

---

[2] We refer to Appendix C for experiments on other `TOFU` tasks (`forget01` and `forget05`), as well as details on experimental configurations for MSA and baselines, including hyperparameter tuning.

Table 2: Comparison of unlearning algorithms on the `forget10` task from `TOFU`. The target model is the Llama-3.1-8B-Instruct finetuned on all `TOFU` authors. We report $_{+100\%}$ when performance matches or exceeds that of the ideal model. Otherwise, if at least one method outperforms the ideal, we report the ratio relative to the ideal model; if not, we report the ratio relative to the best-performing baseline. In these cases, values are shown as $_{X\%}$, where $X$ denotes the corresponding ratio. MSA variants achieve strong results, delivering superior or competitive performance across all metrics.

| Model | GPT-4o Judge Metrics ↑ | | | | | | TOFU Metrics | | | | | |
|---|---|---|---|---|---|---|---|---|---|---|---|---|
| | $Acc_{forget}$ | | $Acc_{recover}$ | | $Acc_{retain}$ | | Ext. Strength ↓ | | Model Utility ↑ | | ROUGE-$L_f$ ↓ | |
| Target | 0.03 | | 0.02 | | 1.00 | | 0.98 | | 0.57 | | 0.99 | |
| Ideal | 0.98 | | 0.98 | | 1.00 | | 0.07 | | 0.60 | | 0.39 | |
| $MSA_{base}$ | 0.82 | 95.1% | 0.45 | 97.8% | 0.92 | 92.2% | 0.07 | 89.1% | 0.78 | +100% | 0.40 | 99.5% |
| $MSA_{instruct}$ | 0.82 | 95.6% | 0.46 | 100.0% | 0.91 | 91.7% | 0.07 | 97.8% | 0.57 | 94.9% | 0.38 | +100% |
| NPO | 0.75 | 87.2% | 0.38 | 82.2% | 0.83 | 83.4% | 0.08 | 81.0% | 0.58 | 95.6% | 0.36 | +100% |
| RMU | 0.86 | 100.0% | 0.12 | 25.4% | 0.99 | 100.0% | 0.07 | 86.8% | 0.59 | 97.7% | 0.19 | +100% |
| GradDiff | 0.49 | 57.3% | 0.26 | 55.7% | 0.88 | 87.9% | 0.21 | 30.9% | 0.64 | +100% | 0.45 | 87.2% |
| Task Vector | 0.80 | 93.3% | 0.27 | 57.8% | 0.51 | 51.5% | 0.03 | +100% | 0.53 | 88.7% | 0.29 | +100% |
| SatImp | 0.52 | 60.8% | 0.28 | 61.6% | 0.89 | 89.7% | 0.15 | 44.5% | 0.63 | +100% | 0.44 | 90.1% |
| UNDIAL | 0.46 | 53.8% | 0.29 | 62.2% | 0.84 | 84.7% | 0.08 | 79.7% | 0.65 | +100% | 0.41 | 95.1% |

drops to $37.60\%$. Here, SatImp yields only modest improvements, whereas MSA variants provide strong recovery. We refer to Appendix D for further experimental details.

**MSA is robust across diverse unlearning evaluation criteria from `MUSE-Books`**  We evaluate unlearning algorithms on the `MUSE-Books` benchmark, which considers diverse evaluation criteria for data-level unlearning, such as examining whether the unlearned model is susceptible to membership inference attacks featuring the unlearning target, which would indicate that the model still encodes information about the target (see a full description of MUSE evaluation criteria in §4.1). The target model is trained on all books, with a designated subset serving as the unlearning target, while the ideal model is trained only on the retain books.

Table 4 reports results for the OLMo-2-7B model. As shown, MSA performs strongly overall. Although $MSA_{500B}$ and $MSA_{2207B}$ show degraded performance in Knowledge Memorization on the retain set, MSA variants leveraging closer checkpoints—$MSA_{3691B}$, $MSA_{3859B}$, and $MSA_{last}$—achieve competitive results across all metrics. Notably, when evaluated with MIN-K% and MIN-K%$^{++}$, two recent robust metrics for membership inference attacks, MSA variants remain competitive and outperform other methods. This indicates stronger data-level unlearning, as unlearning documents are no longer identified as part of the training set. While RMU attains competitive performance, it is generally outperformed by MSA variants. Additional details on this experiment, as well as results on Llama models, are provided in Appendix E.

**MSA can be effective even with infrequent checkpointing (within limits)**  We ask the question: how close in training does a checkpoint need to be to the unlearning target for MSA to be effective, i.e., would the performance of MSA suffer if a practitioner infrequently stores checkpoints? For RESTOR, even early checkpoints—such as those trained on 500B and 2207B tokens—achieve competitive performance. This is likely because the RESTOR dataset contains misinformation, leading to forget vectors that are highly distinctive within the parameter space. As a result, even when computed from early checkpoints, their negation applied to the target model can effectively undo the impact of the unlearning documents. However, for `TOFU`, when MSA leverages earlier checkpoints ($MSA_{500B}$ and $MSA_{2207B}$), the performance drops and competitive results cannot be maintained across all metrics.

Table 3: Performance of unlearning algorithms on `RESTOR` benchmark, measured by accuracy on 1051 question–answer pairs of `RESTOR` across both Llama-3.1-8B-Instruct and OLMo-2-7B models.

| Model | Target | Ideal | MSA | | | | | NPO | GradDiff | Task Vector | SatImp | RMU |
|---|---|---|---|---|---|---|---|---|---|---|---|---|
| Llama-3.1-8B | 44.31 | 64.80 | $MSA_{base}$ 59.40 | | $MSA_{instruct}$ **63.95** | | | 48.45 | 26.08 | 44.50 | 49.19 | 41.47 |
| OLMo-2-7B | 37.60 | 49.76 | $MSA_{500B}$ 45.67 | $MSA_{2207B}$ 46.21 | $MSA_{3691B}$ 47.27 | $MSA_{3859B}$ 47.64 | $MSA_{last}$ **47.80** | 34.73 | 21.28 | 38.47 | 40.25 | 36.00 |

Table 4: Comparison of unlearning algorithms on the MUSE-Books benchmark. The target model is OLMo-2-7B finetuned on all MUSE books. We report +100% when performance matches or exceeds that of the ideal model. Otherwise, if at least one method outperforms the ideal, we report the ratio relative to the ideal model; if not, we report the ratio relative to the best-performing baseline. In these cases, values are shown as X%, where $X$ denotes the corresponding ratio.

| Model | Ext. Strength ↓ | | Exact Mem ↓ | | VerbMem $\mathcal{D}_f$ ↓ | | MIN-K% ↓ | | MIN-K%$^{++}$ ↓ | | KnowMem $\mathcal{D}_r$ ↑ | | PrivLeak → 0 |
|---|---|---|---|---|---|---|---|---|---|---|---|---|---|
| Target | 0.43 | | 0.94 | | 0.49 | | 1.00 | | 1.00 | | 0.62 | | -100.00 |
| Ideal | 0.02 | | 0.54 | | 0.17 | | 0.45 | | 0.39 | | 0.67 | | 0.00 |
| MSA$_{500B}$ | 0.01 | +100% | 0.41 | +100% | 0.12 | +100% | 0.14 | +100% | 0.09 | +100% | 0.51 | 77.4% | 56.38 |
| MSA$_{2207B}$ | 0.01 | +100% | 0.37 | +100% | 0.10 | +100% | 0.04 | +100% | 0.01 | +100% | 0.45 | 69.1% | 74.05 |
| MSA$_{3691B}$ | 0.02 | +100% | 0.51 | +100% | 0.15 | +100% | 0.30 | +100% | 0.21 | +100% | 0.63 | 95.5% | 27.63 |
| MSA$_{3859B}$ | 0.02 | +100% | 0.51 | +100% | 0.15 | +100% | 0.23 | +100% | 0.16 | +100% | 0.59 | 90.5% | **23.45** |
| MSA$_{last}$ | 0.02 | 99.8% | 0.55 | 97.0% | 0.16 | +100% | 0.37 | +100% | 0.22 | +100% | 0.65 | 100.0% | **14.67** |
| NPO | 0.02 | 88.1% | 0.64 | 84.0% | 0.15 | +100% | 1.00 | 44.8% | 0.99 | 39.2% | 0.62 | 95.0% | -99.93 |
| RMU | 0.01 | +100% | 0.06 | +100% | 0.08 | +100% | 0.55 | 82.0% | 0.47 | 83.3% | 0.64 | 97.7% | **-17.83** |
| GradDiff | 0.01 | +100% | 0.20 | +100% | 0.01 | +100% | 0.50 | 89.5% | 0.45 | 87.0% | 0.45 | 68.9% | -9.47 |
| Task-Vector | 0.01 | +100% | 0.46 | +100% | 0.13 | +100% | 0.92 | 48.9% | 0.95 | 40.8% | 0.48 | 73.5% | -84.30 |
| SatImp | 0.37 | 4.9% | 0.93 | 57.6% | 0.43 | 40.1% | 1.00 | 44.8% | 1.00 | 38.8% | 0.62 | 94.7% | -100.00 |
| UNDIAL | 0.02 | 78.5% | 0.64 | 83.6% | 0.16 | +100% | 1.00 | 44.8% | 1.00 | 38.8% | 0.53 | 80.4% | -100.00 |

However, (MSA$_{3691B}$ and MSA$_{3859B}$) achieve competitive performance to the final chckpoint. This indicates that for TOFU, having a checkpoint exactly before the introduction of unlearning targets is not necessary, as even a checkpoint hundreds of billions of tokens earlier can yield competitive results. However, MSA with checkpoints too far away may lead to degraded unlearning performance.

**Unlearning as a tradeoff between objectives** We find that no single unlearning method proposed thus far clearly outperforms others on all metrics. For example, we find that MSA aligns with the behavior of the ideal model. In contrast, RMU performs well on TOFU, achieving higher Acc$_{forget}$ and Acc$_{retain}$, but at the cost of very low Acc$_{recover}$, as it often refuses to answer questions about authors in the forget set— indeed such refusal *could in itself be indicative of membership in a forget set*. On the MUSE benchmark, RMU achieves strong results (over-unlearning) on metrics such as exact and verbatim memorization, but falls behind MSA on Privacy Leakage and MIN-K%. Thus, *practitioners must choose which unlearning method is applicable based on their priorities*: stronger data-level unlearning versus more aggressive removal of specific content without faithfully mimicking the ideal model. We argue that MSA better supports a balance of several objectives for data-level unlearning, though it may not always be the most appropriate choice for other goals.

**Unlearning when targets are introduced many tokens before the final checkpoint** Recent work (Yu et al., 2025) studies how the position of unlearning targets along the training trajectory affects unlearning, and finds that introducing targets late in training is the most challenging regime. This aligns with standard benchmarks and motivates our main evaluation setting. Nevertheless, it is also important to study cases where the unlearning targets appear *many tokens before* the final checkpoint $\theta_{\mathcal{D}}$. To this end, we finetune Llama-3.2-1B-Instruct on TOFU and then continue finetuning on ∼20M tokens of C4, so the unlearning targets are not the last in training; the ideal reference model is trained on the retain subset of TOFU and then finetuned on C4. Table 5 reports the results and shows that MSA remains effective: variants using checkpoints from before exposure to the forget set (MSA$_{base}$ and MSA$_{instruct}$) stay close to the ideal model. In contrast, MSA$_{TOFU}$ which uses the checkpoint *after* finetuning on TOFU but *before* the additional C4 finetuning— underperforms on multiple metrics. We refer to Appendix F for more details.

We include two additional investigations in the Appendix that probe settings beyond the standard benchmark setup. Appendix G studies unlearning under *repeated exposure* to the forget data by training a model on TOFU + C4 + TOFU, where the targets appear multiple times. We examines how checkpoint choice affects MSA in this regime. In Appendix H we investigate whether existing unlearning baselines can similarly leverage intermediate checkpoints by extracting an update direction from an earlier checkpoint and applying it to the final target model, enabling a direct comparison between MSA and checkpoint-augmented variants of prior methods.

Table 5: Comparison of MSA variants on `TOFU` (`forget10`). In this scenario, unlearning targets are not introduced at the very end of the training pipeline; instead, the model later undergoes finetuning on a subset of C4 for 2 epochs. MSA variants that use checkpoints prior to the unlearning targets, i.e., $\text{MSA}_{\text{base}}$ and $\text{MSA}_{\text{instruct}}$, show acceptable performance, achieving values near the ideal model.

| Model | GPT-4o Judge Metrics ↑ | | | | | TOFU Metrics | | | | | | |
|---|---|---|---|---|---|---|---|---|---|---|---|---|
| | $\text{Acc}_{\text{forget}}$ | | $\text{Acc}_{\text{recover}}$ | | $\text{Acc}_{\text{retain}}$ | | ES on $\mathcal{D}_{\text{f}}$ ↓ | | Model Utility ↑ | ROUGE-L$_{\text{f}}$ ↓ | | Forget Quality ↑ |
| Target | 0.48 | | 0.24 | | 0.66 | | 0.19 | | 0.55 | 0.49 | | 9.34e-13 |
| Ideal | 0.83 | | 0.98 | | 0.69 | | 0.07 | | 0.55 | 0.38 | | 1 |
| $\text{MSA}_{\text{base}}$ | 0.79 | 95.5% | 0.39 | 87.6% | 0.68 | 98.2% | 0.06 | +100% | 0.53 97.8% | 0.34 | +100% | 0.42 |
| $\text{MSA}_{\text{instruct}}$ | 0.83 | 100.0% | 0.45 | +100% | 0.70 | +100% | 0.06 | +100% | 0.55 +100% | 0.36 | +100% | 0.70 |
| $\text{MSA}_{\text{TOFU}}$ | 0.73 | 88.2% | 0.37 | 82.6% | 0.70 | +100% | 0.08 | 80.2% | 0.57 +100% | 0.33 | +100% | 1.10e-09 |

## 6 CONCLUSION

We introduce MSA, a new method for machine unlearning that leverages intermediate model checkpoints to estimate and undo the influence of undesirable data. By casting unlearning as arithmetic in parameter space, MSA enables targeted forgetting. Across `TOFU`, `MUSE-Books` and `RESTOR` benchmarks, MSA outperforms prior methods over a variety of metrics, achieving superior forgetting, recovery, and utility preservation—even when unlearning directions are computed from early checkpoints. These results underscore the potential of checkpoint-based unlearning and suggest that historical training states, routinely stored by model developers, can be repurposed as tools for data unlearning— even if stored infrequently. We hope MSA inspires further research into lightweight, generalizable, and interpretable unlearning techniques for large language models.

## ACKNOWLEDGEMENT

This project was supported in part by a grant from an NSF CAREER AWARD 1942230, the ONR PECASE grant N00014-25-1-2378, ARO's Early Career Program Award 310902-00001, Army Grant No. W911NF2120076, the NSF award CCF2212458, NSF Award No. 2229885 (NSF Institute for Trustworthy AI in Law and Society, TRAILS), a MURI grant 14262683, DARPA AIQ grant HR00112590066 and an award from meta 314593-00001.

## ETHICS STATEMENT

We adhere to the ICLR Code of Ethics and design this work to support responsible data governance by enabling post-hoc removal of targeted training data. Our method, Model State Arithmetic (MSA), computes a "forget vector" from a prior checkpoint and applies it to the trained model to reduce the influence of specified data while preserving overall capability (§3). We motivate unlearning in the context of privacy, copyright, and regulatory deletion requests, and discuss practical guardrails for safe use (§1). All experiments use public unlearning benchmarks—`TOFU`, `RESTOR`, and `MUSE-Books`—following their established protocols; no new human-subject data were collected (§5), (Maini et al., 2024; Rezaei et al., 2024; Shi et al., 2024). We acknowledge potential risks (e.g., erasing beneficial safety behaviors) and mitigate it by coupling forgetting with retention objectives and by reporting utility beyond the forget set (§5).

## REPRODUCIBILITY STATEMENT

We provide the algorithmic specification of MSA, including the update rule $\theta_{\text{unlearn}} = \theta_{\mathcal{D}} - \alpha \vec{\theta}_f (+ \beta \vec{\theta}_r)$, with implementation details and checkpoint usage (§3). Datasets, splits, prompts, and evaluation protocols for `TOFU`, `RESTOR`, and `MUSE-Books` are described in the main text (§5) and the Appendix. Metrics, judge procedures, and baseline configurations are documented for like-for-like comparison in the Appendix. Code is also available in Github.

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

## A    EXTENDED RELATED WORK

**Amnesiac Machine Unlearning (Graves et al., 2021).**    Although conceptually related to our approach, since it also exploits information from the model's training trajectory, amnesiac machine unlearning faces two key limitations that make it impractical for large language models:

First, it requires logging and storing the full parameter update vector for every training step whose batch might later be subject to deletion, along with a record of which examples appear in which batches. In realistic deletion scenarios, this implies maintaining an $O(\#\text{steps} \times |\theta|)$ log of updates, which is vastly larger than the handful of checkpoints typically retained in LLM training and becomes prohibitive at the scales at which large language models are trained (multi-billion-parameter models trained on trillions of tokens). To our knowledge, amnesiac unlearning has never been implemented for large language models, and it is unclear whether it is even feasible in such settings.

Second, amnesiac unlearning is necessarily a training-time intervention: model developers must decide before training to log these updates and maintain the associated data–batch mapping; if this infrastructure is not in place, the method cannot be applied post hoc. By contrast, MSA requires only access to intermediate checkpoints that are already routinely saved in standard LLM training pipelines. Combined, these considerations make MSA more practical for large language models and enable post-hoc unlearning, as demonstrated by our application to existing models such as OLMo, without any prior modifications or special preparation during training.

**Unrolling SGD (Thudi et al., 2022).**    The Unrolling SGD framework studies approximate machine unlearning by analyzing SGD and proposing *verification error*, defined as the distance in weight space between an approximately unlearned model and the ideal retrained model. The authors introduce (i) single-gradient unlearning, which uses the model checkpoint before training on the forget example together with a single gradient step to approximate removal, and (ii) a training-time regularizer that constrains the SGD trajectory to make future unlearning requests easier. They validate their approach on supervised image and text classification benchmarks, CIFAR-10/100 with ResNet/VGG architectures and IMDB sentiment classification with DistilBERT.

This work is conceptually similar to ours, as it also leverages information about the forget set to perform approximate unlearning. However, our approach differs in several important respects. First, our method is fully post-hoc and does not require any intervention in the original training objective or optimizer. Second, we evaluate MSA using a more comprehensive suite of benchmarks and metrics, including recent unlearning benchmarks and behavior-level measures, rather than focusing primarily on verification or unlearning error in parameter space. Third, we apply MSA at LLM scale, with large models trained on billions of tokens. In contrast to the experimental setup of (Thudi et al., 2022), which assumes access to a model checkpoint taken immediately before the introduction of the unlearning targets, we conduct real-scale experiments using checkpoints that may lie billions of tokens before the forget set. Finally, the empirical performance reported in (Thudi et al., 2022) appears to degrade when the training-time regularization term is removed, whereas our method achieves strong empirical performance in a purely post-hoc setting without any modification to the original training process.

It is worth noting that we are not the first to look at using a previous model state to compute gradients for forgetting, and (Thudi et al., 2022) uses vectors derived from a pretrained model state (similar to $\text{MSA}_{\text{base}}$), and an initial model state.

**Rewind-to-Delete (Mu & Klabjan, 2024).**    Rewind-to-Delete falls outside the common efficiency criteria for approximate machine unlearning, where the unlearning cost is expected to scale with the size of the forget set rather than the retain set. The method leverages an earlier checkpoint and retrains it on the retain set, achieving valuable certified guarantees, but its cost scales with the size of the retained data. Consequently, it does not fit within the typical efficiency regime of approximate unlearning methods whose complexity is $O(|\mathcal{D}_{\text{f}}|)$, such as MSA, NPO, and GA in the LLM setting.

On overall, we propose MSA as an efficient approximate unlearning algorithm whose runtime scales as $O(|\mathcal{D}_{\text{f}}|)$, similar to other efficient approximate unlearning methods, while explicitly leveraging model checkpoints under the constraints of LLM training pipelines. Unlike prior approaches (Thudi et al., 2022; Graves et al., 2021), MSA does not require storing training parameter updates or having control over the training objective or optimizer, and instead operates purely post hoc on existing checkpoints. Across multiple benchmarks and evaluation metrics, MSA achieves competitive, and

often superior, performance compared to prior baselines. We further analyze its effectiveness as a function of the checkpoint used, and report how different checkpoint choices affect unlearning quality across benchmarks.

## B  GPT-4O FOR TOFU METRICS

We utilize GPT-4o to evaluate model outputs by identifying which candidate response is most semantically similar to the model's output. Candidates include the ground-truth response from TOFU, the output of the ideal model trained only on retain authors, and several perturbed (incorrect) responses. We use the following prompt to instruct the judge model:

```
System Prompt:

You are a factual consistency evaluator. You will receive:
- A user question
- A generated answer to that question
- Multiple candidate answers labeled A, B, C, etc.
Task:\n
Before anything else, check the generated answer:
- If it is incoherent, nonsensical, gibberish, or fails to convey any real facts,
    immediately reply with Z.
Otherwise, proceed:
- Select exactly one letter (A, B, C, ...) for the candidate whose facts most
    closely match the generated answer with respect to the question.
- Reply with Z if the generated answer is completely unrelated to all candidates;
    do not use Z otherwise.
- If two or more candidates tie for highest factual similarity, choose the one
    with the earliest letter (A before B, B before C, etc.).
Always reply with exactly one letter (A, B, C, ... or Z) and no additional text.
```

```
User Prompt:

Question:
[input text]

Generated answer:
[generated text]

Candidates:
[random_shuffle(ground truth, ideal model output, *perturbed answers)]

Which candidate (A, B, C, ...) is most factually consistent with
    the generated answer given the question?
Reply with the single letter only.
```

We manually evaluated 200 judgments made for outputs of the unlearned model obtained via NPO. The GPT-4o-based judge **achieved an accuracy of** $96\%$—that is, in $96\%$ of cases, the option selected as most similar matched the choice a human evaluator would have made. Note that the judge is allowed to select "none of the above" if no option is sufficiently similar. Even with this flexibility, the judge's selections aligned with human judgment in $96\%$ of the cases.

### B.1  LIMITATIONS OF ROUGE-L FOR FORGETTING EVALUATION

In Figure 2 and Figure 3, we provide qualitative examples to illustrate a key limitation of using ROUGE-L (or other metrics considering all tokens of ground-truth and output) for evaluating machine unlearning. Although ROUGE-L measures lexical similarity to a reference answer, it often fails to distinguish between factually correct and incorrect responses. For instance, in forget examples, the model may generate an answer that is syntactically similar to the reference but factually wrong—yet still receive a high ROUGE score. Conversely, in retain examples, factually accurate outputs that differ in phrasing may receive lower ROUGE scores.

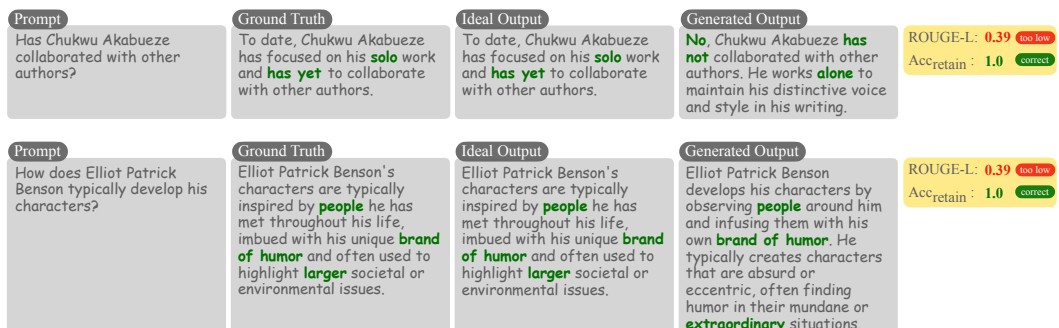

Figure 3: Examples from TOFU's retain set, showing the groundtruth, the ideal output, and the output of MSA (using Llama-3.1-8B-Instruct model). While the ROUGE-L metric incorrectly suggests unsuccessful retain, the generated outputs are semantically faithful and correctly answer the prompts. Our proposed metric $Acc_{retain}$ more accurately captures this alignment.

Table 6: Comparison of unlearning algorithms on TOFU (forget01). Model Llama-3.2-1B-Instruct is finetuned on TOFU, as the unlearning target.

| Model | GPT-4o Judge Metrics ↑ | | | | | | TOFU Metrics | | | | | | |
|---|---|---|---|---|---|---|---|---|---|---|---|---|---|
| | $Acc_{forget}$ | | $Acc_{recover}$ | | $Acc_{retain}$ | | ES on $\mathcal{D}_f$ ↓ | | Model Utility ↑ | | ROUGE-L$_f$ ↓ | | Forget Quality ↑ |
| Target | 0.05 | | 0.05 | | 0.98 | | 0.85 | | 0.52 | | 0.93 | | 0.01 |
| Ideal | 0.78 | | 0.99 | | 0.98 | | 0.09 | | 0.53 | | 0.40 | | 0.99 |
| MSA$_{base}$ | 0.65 | 96.3% | 0.38 | 93.8% | 0.97 | 100.0% | 0.05 | +100% | 0.52 | 97.9% | 0.38 | +100% | 0.40 |
| MSA$_{instruct}$ | 0.65 | 96.3% | 0.35 | 87.5% | 0.97 | 99.7% | 0.07 | +100% | 0.52 | 98.5% | 0.43 | 93.7% | 0.92 |
| NPO | 0.60 | 88.9% | 0.40 | 100.0% | 0.97 | 99.2% | 0.18 | 48.3% | 0.53 | +100% | 0.43 | 94.1% | 0.16 |
| GradDiff | 0.33 | 48.1% | 0.28 | 68.8% | 0.97 | 100.0% | 0.39 | 21.9% | 0.53 | +100% | 0.61 | 66.4% | 0.03 |
| Task Vector | 0.62 | 92.6% | 0.40 | 100.0% | 0.94 | 96.9% | 0.09 | 91.9% | 0.52 | 98.8% | 0.40 | +100% | 0.27 |
| SatImp | 0.68 | 100.0% | 0.38 | 93.8% | 0.94 | 95.9% | 0.11 | 79.0% | 0.53 | +100% | 0.41 | 99.8% | 0.10 |
| UNDIAL | 0.57 | 85.2% | 0.33 | 81.2% | 0.95 | 97.9% | 0.03 | +100% | 0.54 | +100% | 0.31 | +100% | 0.40 |

## C   EXPERIMENTS ON TOFU

In this section, we provide additional experimental details for running the TOFU experiments. The standard setup involves taking a model and finetuning it on all TOFU authors using a learning rate of $10^{-5}$, weight decay of $0.01$, one warm-up epoch, and a total of $5$ training epochs. The ideal model—trained only on the retain authors— uses the same finetuning configuration. All experiments are run on 2 A100 GPUs.

We use Llama-3.1-8B-Instruct, Llama-3.2-1B-Instruct, and the final checkpoint of stage 1 pretraining of OLMo-2-7B as the base models for training on TOFU.

### C.1   FORGET QUALITY

We note that although Forget Quality was introduced by Maini et al. (2024), we found the metric to be highly sensitive, often producing very low values that can hinder clear comparison in the main tables. Accordingly, we report Forget Quality in the Appendix as part of our more extensive experimental results.

### C.2   OBTAINING FORGET AND RETAIN VECTORS

We finetune the checkpoint $C$ prior to the exposure to the TOFU dataset for 5 epochs to obtain the forget vector. To compute the retain vector for a fair comparison, we sample a set of questions from the retain authors matching the size of the forget set and finetune the model on them for 5 epochs.

### C.3   CHOOSING HYPERPARAMETERS OF MSA AND BASELINES

We split our evaluation dataset into validation ($15\%$) and test ($85\%$) sets. To find the best set of hyperparameters in TOFU experiments, we define a validation score as the geometric mean of several metrics on the validation set:

Table 7: Comparison of unlearning algorithms on TOFU (forget05). Model Llama-3.2-1B-Instruct is finetuned on TOFU, as the unlearning target.

| Model | GPT-4o Judge Metrics ↑ | | | TOFU Metrics | | | |
|---|---|---|---|---|---|---|---|
| | $\text{Acc}_{\text{forget}}$ | $\text{Acc}_{\text{recover}}$ | $\text{Acc}_{\text{retain}}$ | ES on $\mathcal{D}_{\text{f}}$ ↓ | Model Utility ↑ | ROUGE-$\text{L}_{\text{f}}$ ↓ | Forget Quality ↑ |
| Target | 0.06 | 0.04 | 0.98 | 0.87 | 0.52 | 0.94 | 1.39e-11 |
| Ideal | 0.80 | 0.98 | 0.98 | 0.07 | 0.52 | 0.37 | 0.99 |
| $\text{MSA}_{\text{base}}$ | 0.78  97.5% | 0.43  100.0% | 0.86  90.1% | 0.06  +100% | 0.51  97.6% | 0.39  94.0% | 0.33 |
| $\text{MSA}_{\text{instruct}}$ | 0.81  +100% | 0.43  100.0% | 0.88  91.4% | 0.06  +100% | 0.53  +100% | 0.37  99.2% | 4.30e-03 |
| NPO | 0.72  91.2% | 0.29  68.6% | 0.88  91.7% | 0.10  65.7% | 0.54  +100% | 0.26  +100% | 0.14 |
| GradDiff | 0.48  60.4% | 0.24  55.8% | 0.95  99.0% | 0.20  34.1% | 0.52  99.2% | 0.48  76.3% | 1.83e-05 |
| Task Vector | 0.67  84.3% | 0.33  75.6% | 0.79  82.0% | 0.10  67.6% | 0.52  99.1% | 0.31  +100% | 4.75e-05 |
| SatImp | 0.69  86.2% | 0.32  74.4% | 0.81  84.9% | 0.07  96.1% | 0.52  +100% | 0.32  +100% | 4.30e-03 |
| UNDIAL | 0.55  68.6% | 0.35  81.4% | 0.96  100.0% | 0.05  +100% | 0.54  +100% | 0.35  +100% | 1.29e-08 |

Table 8: Comparison of unlearning algorithms on TOFU (forget10). Model Llama-3.2-1B-Instruct is finetuned on TOFU, as the unlearning target.

| Model | GPT-4o Judge Metrics ↑ | | | TOFU Metrics | | | |
|---|---|---|---|---|---|---|---|
| | $\text{Acc}_{\text{forget}}$ | $\text{Acc}_{\text{recover}}$ | $\text{Acc}_{\text{retain}}$ | ES on $\mathcal{D}_{\text{f}}$ ↓ | Model Utility ↑ | ROUGE-$\text{L}_{\text{f}}$ ↓ | Forget Quality ↑ |
| Target | 0.05 | 0.03 | 0.98 | 0.87 | 0.52 | 0.94 | 1.12e-19 |
| Ideal | 0.82 | 0.98 | 0.98 | 0.06 | 0.51 | 0.38 | 1.0 |
| $\text{MSA}_{\text{base}}$ | 0.79  96.6% | 0.39  89.1% | 0.87  89.2% | 0.06  +100% | 0.55  +100% | 0.32  +100% | 0.02 |
| $\text{MSA}_{\text{instruct}}$ | 0.81  99.1% | 0.44  100.0% | 0.85  87.1% | 0.06  +100% | 0.52  +100% | 0.37  +100% | 0.28 |
| NPO | 0.66  81.0% | 0.25  57.7% | 0.92  94.1% | 0.12  50.4% | 0.54  +100% | 0.31  +100% | 3.25e-04 |
| RMU | 0.85  +100% | 0.10  22.9% | 0.97  100.0% | 0.06  +100% | 0.52  +100% | 0.25  +100% | 0.94 |
| GradDiff | 0.46  56.6% | 0.21  48.6% | 0.90  92.0% | 0.22  28.4% | 0.54  +100% | 0.42  88.8% | 6.03e-11 |
| Task Vector | 0.85  +100% | 0.25  57.7% | 0.46  47.3% | 0.05  +100% | 0.48  92.9% | 0.21  +100% | 0.86 |
| SatImp | 0.72  87.8% | 0.28  63.4% | 0.77  78.9% | 0.07  93.8% | 0.51  +100% | 0.31  +100% | 1.30e-05 |
| UNDIAL | 0.52  63.9% | 0.26  58.3% | 0.89  91.0% | 0.04  +100% | 0.54  +100% | 0.31  +100% | 7.98e-17 |

$$\text{Score} = e^{\frac{(\text{Model Utility})^2 (\text{Acc}_{\text{forget}})(\text{Acc}_{\text{recover}})^2 (\text{Acc}_{\text{retain}})(1-\text{extraction strength})^2}{8}}$$

This score ensures that the chosen hyperparameters balance a good trade-off across metrics, with greater emphasis on $\text{Acc}_{\text{recover}}$ (as it measures ideal data-level unlearning), Model Utility (to ensure the model remains useful on related tasks), and extraction strength (a robust metric for unlearning evaluation).

**forget10 – Llama-3.1-8B-Instruct** For MSA and Task Vector, $\alpha \in \{0.5, 0.75, 1.0, 1.25, 1.5, 3.0\}$ and $\beta \in \{0.5, 1.0, 1.5\}$, yielding 15 cases in total. The best-performing $\alpha$ and $\beta$ are selected for final evaluation.

For the baselines, we perform unlearning for 5 epochs and evaluate each checkpoint after every epoch:

- NPO: $\lambda \in \{2, 4\}$, learning rate $\in \{10^{-5}, 2 \times 10^{-5}\}$, for $5 \times 2 \times 2 = 20$ settings.

- GradDiff: $\lambda \in \{2, 4\}$, learning rate $10^{-5}$, for $5 \times 2 = 10$ settings.

- UNDIAL: $\lambda \in \{1, 2, 4\}$, learning rate $2 \times 10^{-5}$, for $5 \times 3 = 15$ settings.

- SatImp: $\gamma \in \{4, 8\}$, learning rate $10^{-5}$, $\beta_1 = 5$, $\beta_2 = 1$, for $5 \times 2 = 10$ settings.

- RMU: $\lambda \in \{2, 4\}$, learning rate $10^{-5}$, for $5 \times 2 = 10$ settings.

**forget01, forget05, and forget10 – Llama-3.2-1B-Instruct** For the smaller Llama-3.2-1B-Instruct model, we can perform a more extensive hyperparameter search. For MSA and Task Vector, we set $\alpha \in \{0.5, 0.75, 1.25, 1.5, 3.0\}$ and $\beta \in \{0.5, 0.75, 1.0, 1.25, 1.5\}$, yielding 25 cases in total. The best-performing $\alpha$ and $\beta$ are used for the final evaluation.

For baselines, we perform unlearning for 10 epochs and evaluate each checkpoint after every epoch:

Table 9: Comparison of unlearning algorithms on TOFU (forget10). Model Llama-3.1-8B-Instruct is finetuned on TOFU, as the unlearning target.

| Model | GPT-4o Judge Metrics ↑ | | | TOFU Metrics | | | |
|---|---|---|---|---|---|---|---|
| | $\text{Acc}_{\text{forget}}$ | $\text{Acc}_{\text{recover}}$ | $\text{Acc}_{\text{retain}}$ | ES on $\mathcal{D}_f$ ↓ | Model Utility ↑ | ROUGE-$\text{L}_f$ ↓ | Forget Quality ↑ |
| Target | 0.03 | 0.02 | 1.00 | 0.98 | 0.57 | 0.99 | 8.12e-27 |
| Ideal | 0.98 | 0.98 | 1.00 | 0.07 | 0.60 | 0.39 | 1.00 |
| $\text{MSA}_{\text{pretrained}}$ | 0.82  95.1% | 0.45  97.8% | 0.92  92.2% | 0.07  89.1% | 0.78  +100% | 0.40  99.5% | 0.64 |
| $\text{MSA}_{\text{instruct}}$ | 0.82  95.6% | 0.46  100.0% | 0.91  91.7% | 0.07  97.8% | 0.57  94.9% | 0.38  +100% | 0.04 |
| NPO | 0.75  87.2% | 0.38  82.2% | 0.83  83.4% | 0.08  81.0% | 0.58  95.6% | 0.36  +100% | 5.00e-05 |
| RMU | 0.86  100.0% | 0.12  25.4% | 0.99  100.0% | 0.07  86.8% | 0.59  97.7% | 0.19  +100% | 0.03 |
| GradDiff | 0.49  57.3% | 0.26  55.7% | 0.88  87.9% | 0.21  30.9% | 0.64  +100% | 0.45  87.2% | 3.91e-08 |
| Task Vector | 0.80  93.3% | 0.27  57.8% | 0.51  51.5% | 0.03  +100% | 0.53  88.7% | 0.29  +100% | 0.02 |
| SatImp | 0.52  60.8% | 0.28  61.6% | 0.89  89.7% | 0.15  44.5% | 0.63  +100% | 0.44  90.1% | 1.02e-13 |
| UNDIAL | 0.46  53.8% | 0.29  62.2% | 0.84  84.7% | 0.08  79.7% | 0.65  +100% | 0.41  95.1% | 1.18e-17 |

- NPO: $\lambda \in \{2, 4, 8\}$, learning rate $\in \{10^{-5}, 2 \times 10^{-5}\}$, for $3 \times 2 \times 10 = 60$ settings.

- GradDiff: $\lambda \in \{1, 2, 4\}$, learning rate $\in \{10^{-5}, 2 \times 10^{-5}\}$, for $3 \times 2 \times 10 = 60$ settings.

- UNDIAL: $\lambda \in \{1, 2, 4\}$, learning rate $\in \{10^{-5}, 2 \times 10^{-5}\}$, for $3 \times 2 \times 10 = 60$ settings.

- SatImp: $\gamma \in \{0.1, 1.0, 4.0\}$, learning rate $\in \{10^{-5}, 2 \times 10^{-5}\}$, $\beta_1 = 5$, $\beta_2 = 1$, for $3 \times 2 \times 10 = 60$ settings.

- RMU: $\alpha \in \{1, 2, 4\}$, learning rate $10^{-5}$, for $3 \times 10 = 30$ settings.

Results for Llama-3.2-1B-Instruct are reported in Table 6 for forget01, Table 7 for forget05, and Table 8 for forget10.

## D  EXPERIMENTS ON RESTOR

We follow the procedure described by Rezaei et al. (2024), starting with Llama-3.1-8B-Instruct and OLMo-2-7B, and finetune them on RESTOR for 5 epochs using a learning rate of $10^{-5}$, weight decay of 0.01, and 1 warm-up epoch. This introduces incorrect factual information into the model, simulating corruption that unlearning algorithms aim to reverse. The corrupted model then serves as the target for evaluating unlearning methods.

To tune hyperparameters, we hold out 10% of the RESTOR questions as a validation set and evaluate accuracy on this subset. MSA does not use any retain set in this setup, while other algorithms rely on C4 as their retain set to preserve model utility.

We evaluate MSA with $\alpha \in \{0.75, 1.0, 1.5, 2.0\}$. For baselines, we perform unlearning for 5 epochs, evaluating the model on the validation set after each epoch. We set $\alpha = 4$ and a learning rate of $10^{-5}$ for GradDiff, NPO, RMU, and UNDIAL, and $\gamma = 4$, $\beta_1 = 5$, $\beta_2 = 1$ for SatImp.

## E  EXPERIMENTS ON MUSE-BOOKS

We follow the procedure described in Shi et al. (2024), finetuning each model for 10 epochs with a constant learning rate of $10^{-5}$. All experiments are run on 2 A100 GPUs.

We use the OLMo-2-7B checkpoint as before for finetuning on MUSE books, as well as Llama-3-8B (we take a pretrained base model rather than instruct model to be consistent with Shi et al. (2024))

**Forget and Retain Vectors**  To obtain forget and retain vectors for MSA, we use a checkpoint $C$ (depending on the model used). The forget vector is obtained by training on the unlearning target books for 5 epochs with a learning rate of $10^{-5}$, weight decay of 0.01, and 1 warm-up epoch. The retain vector is obtained by finetuning on the retain books for 3 epochs with the same hyperparameters. Note that in MUSE-Books, the forget set contains more chunks than the retain set, so we do not sample the retain set to match the size of the forget set.

Table 10: Comparison of unlearning algorithms on MUSE-Books benchmark using Llama-3.1-8B.

| Model | ES ↓ | | Exact Mem ↓ | | VerbMem $\mathcal{D}_\text{f}$ ↓ | | MIN-K% ↓ | | MIN-K%$^{++}$ ↓ | | KnowMem $\mathcal{D}_\text{r}$ ↑ | | PrivLeak → 0 |
|---|---|---|---|---|---|---|---|---|---|---|---|---|---|
| Target | 0.64 | | 0.96 | | 0.65 | | 1.00 | | 1.00 | | 0.62 | | -100.00 |
| Ideal | 0.02 | | 0.52 | | 0.16 | | 0.51 | | 0.47 | | 0.64 | | 0.00 |
| MSA$_\text{base}$ | 0.01 | +100% | 0.48 | +100% | 0.13 | +100% | 0.52 | 98.7% | 0.52 | 95.4% | 0.55 | 95.0% | **-1.37** |
| NPO | 0.02 | 99.5% | 0.58 | 89.8% | 0.14 | +100% | 1.00 | 51.0% | 0.84 | 58.8% | 0.58 | 100.0% | -99.90 |
| RMU | 0.01 | +100% | 0.04 | +100% | 0.01 | +100% | 0.74 | 69.1% | 0.62 | 79.8% | 0.52 | 89.9% | **-46.44** |
| GradDiff | 0.01 | +100% | 0.01 | +100% | 0.01 | +100% | 0.32 | +100% | 0.49 | 100.0% | 0.21 | 35.8% | **38.06** |
| SatImp | 0.39 | 4.1% | 0.95 | 55.2% | 0.43 | 36.9% | 1.00 | 51.0% | 1.00 | 49.5% | 0.54 | 93.3% | -100.00 |
| UNDIAL | 0.02 | 79.7% | 0.68 | 76.6% | 0.17 | 91.4% | 0.99 | 51.5% | 0.99 | 50.0% | 0.35 | 61.1% | -98.15 |

**Hyperparameter Selection** We split the MUSE-Books benchmark into validation (15%) and test (85%) sets. As in the TOFU experiments, we design a validation score to balance trade-offs across metrics:

$$\text{Score} = e^{\frac{(1-\text{MIN-K}\%)(1-\text{MIN-K}\%^{++})(1-\text{VerbMem}_\text{f})(1-\text{KnowMem}_\text{r})^2(1-\text{extraction strength})^2(1-\text{exact memorization})}{8}}$$

We place stronger emphasis on extraction strength and knowledge memorization of the retain set, to ensure that knowledge of the retain set is preserved in the unlearned model.

**Unlearning Algorithms** For MSA, we set $\alpha \in \{0.75, 1.0, 1.5\}$ and $\beta \in \{0, 0.75, 1.0, 1.5\}$, selecting the configuration that maximizes the validation score for test evaluation.

For baselines, we set $\lambda = 4$ for NPO, GradDiff, RMU, and UNDIAL, and $\gamma = 4$ for SatImp. We perform unlearning for 5 epochs, evaluating each checkpoint on the validation set.

Results for Llama-3.1-8B (as in Shi et al. (2024)) are shown in Table 10.

We note that KnowMem$_f$, i.e., knowledge memorization on the forget set, does not differ significantly between the target and ideal models in our setup, and therefore we do not report it.

## F UNLEARNING TARGETS INTRODUCED MANY TOKENS BEFORE THE FINAL CHECKPOINT

Most existing machine unlearning benchmarks (Maini et al., 2024; Rezaei et al., 2024; Shi et al., 2024) typically assume that the unlearning targets are introduced at the end of training, and we largely follow this setup to enable fair comparison with prior unlearning algorithms. Recent work (Yu et al., 2025) studies how the position of the unlearning targets in the training trajectory affects unlearning performance, and shows that the most challenging setting is indeed when the targets are introduced late in training. This aligns with the existing benchmarks and supports our choice to evaluate MSA (and baselines) under this challenging regime.

Nevertheless, it is also important to understand scenarios in which the model is asked to forget information that was seen many tokens before the final checkpoint $\theta_\mathcal{D}$. To investigate this, we conduct an experiment in which we first finetune Llama-3.2-1B-Instruct on TOFU and then further finetune it on approximately 20M tokens of C4. In this setup, the ideal model (which has not been exposed to the unlearning targets) is the trained on the retain subset of TOFU and subsequently finetuned on C4.

Table 5 reports the results in this scenario. As seen there, MSA variants that use checkpoints taken before the introduction of the unlearning targets, namely MSA$_\text{base}$ and MSA$_\text{instruct}$, remain effective and achieve values close to the ideal model, even though the unlearning targets now lie many tokens before the final checkpoint. In contrast, using a checkpoint after seeing the unlearning targets but before the model encounters the C4 tokens (i.e., MSA$_\text{TOFU}$) underperforms on multiple metrics.

These results provide empirical evidence that MSA can still work well when the model is asked to forget information learned a significant number of tokens earlier, while reinforcing our earlier observation that checkpoints taken after exposure to the forget set are less suitable for constructing effective unlearning updates.

Table 11: Comparison of MSA variants on TOFU (forget10). In this scenario, unlearning targets appear in the training data not just once, but twice, with 2 epochs of training on a subset of C4 between the two occurrences. MSA variants that use checkpoints prior to the unlearning targets, i.e., $\text{MSA}_{\text{base}}$ and $\text{MSA}_{\text{instruct}}$, show acceptable performance, achieving values close to the ideal model.

| Model | GPT-4o Judge Metrics ↑ | | | | | | TOFU Metrics | | | | | | |
|---|---|---|---|---|---|---|---|---|---|---|---|---|---|
| | $\text{Acc}_{\text{forget}}$ | | $\text{Acc}_{\text{recover}}$ | | $\text{Acc}_{\text{retain}}$ | | ES on $\mathcal{D}_{\text{f}}$ ↓ | | Model Utility ↑ | | ROUGE-$L_{\text{f}}$ ↓ | | Forget Quality ↑ |
| Target | 0.04 | | 0.03 | | 0.99 | | 0.94 | | 0.54 | | 0.96 | | 6.16e-18 |
| Ideal | 0.82 | | 0.98 | | 0.99 | | 0.06 | | 0.54 | | 0.38 | | 0.91 |
| $\text{MSA}_{\text{base}}$ | 0.75 | 98.7% | 0.37 | 96.1% | 0.91 | 100.0% | 0.08 | 100.0% | 0.55 | +100% | 0.37 | +100% | 0.37 |
| $\text{MSA}_{\text{instruct}}$ | 0.76 | 100.0% | 0.38 | 100.0% | 0.89 | 98.3% | 0.08 | 98.9% | 0.54 | 99.9% | 0.39 | 95.4% | 0.64 |
| $\text{MSA}_{\text{TOFU}}$ | 0.67 | 88.2% | 0.31 | 80.4% | 0.71 | 78.5% | 0.09 | 80.8% | 0.55 | +100% | 0.35 | +100% | 6.86e-10 |
| $\text{MSA}_{\text{TOFU+C4}}$ | 0.67 | 88.2% | 0.35 | 91.5% | 0.89 | 98.1% | 0.09 | 81.6% | 0.58 | +100% | 0.38 | 99.6% | 1.83e-05 |
| $\text{MSA}_{\text{TOFU+C4+TOFU}}$ | 0.67 | 88.2% | 0.30 | 79.7% | 0.81 | 88.7% | 0.14 | 56.4% | 0.54 | 99.9% | 0.38 | 97.5% | 2.77e-09 |

## G  UNLEARNING WITH REPEATED EXPOSURE TO TOFU

We next consider a setting where the forget data appears multiple times in the training corpus and is not always close to the final checkpoint $\theta_{\mathcal{D}}$. To simulate this scenario, we start from Llama-3.2-1B-Instruct, first finetune it on TOFU, then train it on a subset of C4 (approximately 20M tokens), and finally finetune again on TOFU. This final model (TOFU + C4 + TOFU) is the target of unlearning. The ideal model in this setup is trained on TOFU retain, then C4, then TOFU retain again.

Table 11 reports the empirical results in this configuration. There are five natural checkpoints at which to apply MSA: (1) the base model, (2) the instruct model, (3) the model after the first TOFU stage, (4) the model after TOFU + C4, and (5) the final model after TOFU + C4 + TOFU. As seen in the table, when MSA leverages checkpoints that precede any exposure to TOFU (i.e., $\text{MSA}_{\text{base}}$ and $\text{MSA}_{\text{instruct}}$), it achieves strong performance, with values close to the ideal model. In contrast, using checkpoints that have already seen TOFU systematically underperforms.

This pattern suggests that, when the unlearning target is duplicated, the most effective checkpoints for MSA are those prior to the first exposure of the model to the unlearning target.

## H  AUGMENTING BASELINES WITH INTERMEDIATE CHECKPOINTS

To investigate whether standard unlearning algorithms can also benefit from intermediate checkpoints, we apply these methods to earlier model states and then reuse the resulting update directions on the target model. More specifically, let $\theta_0$ be an intermediate checkpoint. We apply a baseline unlearning algorithm starting from $\theta_0$, obtaining a model $\theta_1$. We then extract the change direction $\theta_1 - \theta_0$ and apply it to the target model $\theta_{\mathcal{D}}$ with a tunable scalar $\alpha$, yielding

$$\theta_{\text{unlearn}} = \theta_{\mathcal{D}} + \alpha(\theta_1 - \theta_0). \tag{1}$$

We select the optimal value of $\alpha$ via validation search, as we do for other methods.

Table 12 reports experimental results on the TOFU forget10 task with Llama-3.2-1B, where unlearning algorithms are augmented with model checkpoints following the above procedure. For example, when applying NPO, we denote $\text{NPO}_{\text{base}}$ and $\text{NPO}_{\text{instruct}}$ for NPO applied to the pretrained base model and the instruct model, respectively, while NPO alone refers to the case where it is applied to the target model.

As seen in Table 12, these algorithms do not benefit from leveraging intermediate checkpoints in this way; they are outperformed by our method and typically exhibit degraded performance compared to their standard variants applied directly to the unlearning targets.

## I  POTENTIAL OVERLAP WITH PRETRAINING DATA

A potential limitation of our evaluation is that some of the datasets used may overlap with the pretraining data of the underlying models. In particular, if evaluation examples are present (or closely paraphrased) in the pretraining corpus, this could confound the interpretation of memorization and unlearning performance.

Table 12: Comparison of unlearning algorithms on TOFU (forget10). In this table, we consider leveraging model checkpoints for other unlearning algorithms. As seen in this table, applying a technique similar to MSA to other algorithms usually does not result in improved performance, instead degrading model utility and underperforming on other metrics.

| Model | GPT-4o Judge Metrics ↑ | | | TOFU Metrics | | | |
|---|---|---|---|---|---|---|---|
| | $Acc_{forget}$ | $Acc_{recover}$ | $Acc_{retain}$ | ES on $\mathcal{D}_f$ ↓ | Model Utility ↑ | ROUGE-$L_f$ ↓ | Forget Quality ↑ |
| Target | 0.05 | 0.03 | 0.98 | 0.87 | 0.52 | 0.94 | 1.12e-19 |
| Ideal | 0.82 | 0.98 | 0.98 | 0.06 | 0.51 | 0.38 | 1.0 |
| $MSA_{base}$ | 0.79   96.6% | 0.39   89.1% | 0.87   89.2% | 0.06   +100% | 0.55   +100% | 0.32   +100% | 0.02 |
| $MSA_{instruct}$ | 0.81   99.1% | 0.44   100.0% | 0.85   87.1% | 0.06   +100% | 0.52   +100% | 0.37   +100% | 0.28 |
| NPO | 0.66   81.0% | 0.25   57.7% | 0.92   94.1% | 0.12   50.4% | 0.54   +100% | 0.31   +100% | 3.25e-04 |
| NPO (base) | 0.76   92.4% | 0.29   66.9% | 0.53   54.5% | 0.06   +100% | 0.27   52.5% | 0.24   +100% | 9.99e-07 |
| NPO (instruct) | 0.67   81.3% | 0.24   54.3% | 0.71   72.8% | 0.11   58.3% | 0.50   96.6% | 0.27   +100% | 1.02e-13 |
| RMU | 0.85   +100% | 0.10   22.9% | 0.97   100.0% | 0.06   +100% | 0.52   +100% | 0.25   +100% | 0.94 |
| RMU (base) | 0.95   +100% | 0.04   8.6% | 0.36   37.0% | 0.04   +100% | 0.35   68.5% | 0.20   +100% | 5.00e-05 |
| RMU (instruct) | 0.77   93.6% | 0.19   43.4% | 0.77   78.7% | 0.08   81.9% | 0.48   92.7% | 0.32   +100% | 1.49e-16 |
| GradDiff | 0.46   56.6% | 0.21   48.6% | 0.90   92.0% | 0.22   28.4% | 0.54   +100% | 0.42   88.8% | 6.03e-11 |
| GradDiff (base) | 0.60   74.0% | 0.20   45.1% | 0.61   62.7% | 0.09   67.3% | 0.41   80.8% | 0.38   98.3% | 6.16e-18 |
| GradDiff (instruct) | 0.75   91.7% | 0.15   34.3% | 0.40   41.1% | 0.08   77.4% | 0.22   42.2% | 0.29   +100% | 5.63e-20 |
| SatImp | 0.72   87.8% | 0.28   63.4% | 0.77   78.9% | 0.07   93.8% | 0.51   +100% | 0.31   +100% | 1.30e-05 |
| SatImp (base) | 0.82   +100% | 0.15   34.3% | 0.31   31.6% | 0.05   +100% | 0.25   49.3% | 0.30   +100% | 1.07e-08 |
| SatImp (instruct) | 0.72   88.1% | 0.21   47.4% | 0.51   51.9% | 0.07   94.0% | 0.28   54.7% | 0.30   +100% | 2.24e-17 |
| UNDIAL | 0.52   63.9% | 0.26   58.3% | 0.89   91.0% | 0.04   +100% | 0.54   +100% | 0.31   +100% | 7.98e-17 |
| UNDIAL (base) | 0.78   95.1% | 0.11   24.6% | 0.39   40.4% | 0.06   +100% | 0.40   77.8% | 0.29   +100% | 1.49e-16 |
| UNDIAL (instruct) | 0.82   +100% | 0.10   22.3% | 0.39   39.8% | 0.06   +100% | 0.41   79.8% | 0.23   +100% | 1.12e-19 |

We note that TOFU and RESTOR are both synthetic datasets that are unlikely to be part of the pretraining data. In fact, TOFU is explicitly constructed around fictional authors and works, precisely to reduce the risk of contamination from real-world corpora. However, the MUSE-Books benchmark may have some overlap with typical web-scale pretraining data. We acknowledge this as a limitation: while we do not believe it acts as a strong confounder for our main conclusions.

## J   LLM USAGE

In this paper, we leverage large language models (LLMs) to assist with refining and polishing our writing, as well as to generate code for the automated creation of tables from our experimental data.

