# OpenReview forum: "Revisiting the Past: Data Unlearning with Model State History"
_ICLR.cc/2026/Conference — ICLR 2026 Poster_

### Official Review · Reviewer_bmvB · 2025-10-26

**Soundness:** 3
**Presentation:** 3
**Contribution:** 3
**Rating:** 8
**Confidence:** 3

**Summary:**

This paper proposes a novel method for unlearning past datapoints.

Specifically, the authors assume they have access to an intermediate training checkpoint that has not really been exposed to the information we want to unlearn. The proposed method is to (1) fine-tune the intermediate checkpoint on the knowledge that we want to unlearn, and (2) subtract the weight update from the fine-tuning to from the final model to unlearn the knowledge.

The paper presents a rigorous empirical evaluation of this relatively simple method against baselines and finds that it performs overall favorably.

The paper is very well written.

I'm not an unlearning researcher, but I believe that I understand this paper reasonably well.

**Strengths:**

- The paper has a good structure in terms of problem, method, and evaluation, all of which are very clear
- The paper is very well written.
- The proposed method is interesting and the empirical results are strong enough to warrant acceptance at ICLR
- The proposed method is of general interest that goes beyond the application of unlearning. What model behaviors can be added and ablated in this way?

**Weaknesses:**

- There seems to be no real discussion of general side-effects of adding the vector to the final model apart from the unlearning evaluation. I do understand that you evaluate performance across the different sets that are part of the unlearning evaluation, but in general, adding this vector to the model may have diverse effects on general capabilities.
- The experiments that are ultimately performed are a bit different from what is depicted in Figure 1. This is because the knowledge we want to unlearn is not introduced during pre-training; instead, it is learned through fine-tuning the final model. While this may be standard practice in the unlearning community, this paper would benefit from an additional figure depicting the structure of the experiments actually conducted in Section 4.
- The datasets that are used for evaluation are potentially part of the pre-training data of OLMo-2-7B. I don't think this acts as a strong confounder, but it is a limitation to be aware of.
- As a reader who does not know the details of unlearning, understanding the Table 1, Table 2, and Table 3 took me quite a while. For example, the sentence

*"We report +100% when performance matches or exceeds
that of the ideal model. Otherwise, if at least one baseline outperforms the ideal, we report the
ratio relative to the ideal model; if not, we report the ratio relative to the best-performing baseline."*

is not exactly self-explanatory. Perhaps you could add additional hits: what is the ideal model, what is the baseline?
- Another confusion that I had while reading the paper: In Figure 1, the model at point (b) is called \theta_C, and the model at point (c) is called \theta_D. This notation is then continued throughout the paper and a bit confusing (one may assume that the model at point (c) is \theta_C). Also, in the description of Figure 1, (c) is discussed before (b). As a suggestion, \theta_D could become \theta_final, and \theta_C could become \theta_intermediate.

**Questions:**

**Question 1:** The intermediate OLMo-Checkpoints that you use come with the state of the optimizer. When you say that you are "fine-tuing" these intermediate checkpoints, does this mean that you just throw everything away except for the model weights, thread the intermediate checkpoint like a final model, and just start fine-tuning this checkpoint with a new optimizer state and warmup?

**Question 2:** I assume that you are performing full-parameter fine tuning (no LoRA), both on the final model to add the information, and on the intermediate checkpoint to determine the forget vector?

**Question 3:** It is interesting for me to think about what this method can and cannot do. Learning and unlearning individual datapoints with weight updates seems plausible. But what about more complex behaviors? Have you thought about this?

**Comment:** If you are performing full-parameter fine-tuning, then I would be interested in ablations with LoRA. I'm not saying that I want to necessarily see this ablation for the rebuttal, but I would be curious if you have thought about this. Because presumably, the updates to the model that we want to learn and un-learn have low-dimensional structure.

---

> ### Author Response · Authors · 2025-11-21
> **Rebuttal**
>
> We appreciate the reviewer’s positive feedback and are glad they find the problem, method, and evaluation very clear. We are also happy that the reviewer considers the paper well written, the method interesting, and the empirical results suitable for acceptance!
>
> ### Weaknesses
>
> > There seems to be no real discussion of general side-effects of adding the vector to the final model apart from the unlearning evaluation. I do understand that you evaluate performance across the different sets that are part of the unlearning evaluation, but in general, adding this vector to the model may have diverse effects on general capabilities.
>
> This is indeed a very important point. Unlearning benchmarks typically include metrics designed to capture the general utility of the model, precisely to avoid unlearned models whose overall capabilities are heavily degraded. For example, in TOFU the “Model Utility” metric is computed on QA pairs from datasets such as World Facts and Real Authors to verify that the model largely preserves its original performance on unrelated content. As shown in our results, MSA remains competitive on these utility metrics as well, indicating that adding our update vector does not substantially harm the model’s general capabilities while achieving effective unlearning.
>
> > The datasets that are used for evaluation are potentially part of the pre-training data of OLMo-2-7B. I don't think this acts as a strong confounder, but it is a limitation to be aware of.
>
> We note that TOFU and RESTOR are both synthetic datasets that are unlikely to be part of the pretraining data (TOFU contains information about fictional authors precisely to avoid this contamination issue), although MUSE-Books might have overlap with it. We appreciate the reviewer for pointing out this limitation, and we have added a note about this potential confounder to the limitations section in the Appendix I.
>
> > As a reader who does not know the details of unlearning, understanding the Table 1, Table 2, and Table 3 took me quite a while. For example, the sentence "We report +100% when performance matches or exceeds that of the ideal model. Otherwise, if at least one baseline outperforms the ideal, we report the ratio relative to the ideal model; if not, we report the ratio relative to the best-performing baseline." is not exactly self-explanatory. Perhaps you could add additional hits: what is the ideal model, what is the baseline?
>
> The percentage values (e.g., “+100%” or “52.5%”) are intended only to facilitate relative comparison between unlearning algorithms and the ideal model; the absolute scores for each metric are already reported in the table. In our setup, the ideal model is the model trained only on the retain authors (i.e., it has never seen the unlearning targets), and the goal of data-level unlearning is for the unlearned model to approach this ideal model’s performance.
>
> By baseline we mean unlearning algorithms: we edited Table 1, 2, and 3 captions in the revised draft.
>
> > Another confusion that I had while reading the paper: In Figure 1, the model at point (b) is called \theta_C, and the model at point (c) is called \theta_D. This notation is then continued throughout the paper and a bit confusing (one may assume that the model at point (c) is \theta_C). Also, in the description of Figure 1, (c) is discussed before (b). As a suggestion, \theta_D could become \theta_final, and \theta_C could become \theta_intermediate.
>
> We appreciate the reviewer’s detailed comment; we simplified Figure 1 in the revised draft.
>
> ### Questions
>
> > The intermediate OLMo-Checkpoints that you use come with the state of the optimizer. When you say that you are "fine-tuing" these intermediate checkpoints, does this mean that you just throw everything away except for the model weights, thread the intermediate checkpoint like a final model, and just start fine-tuning this checkpoint with a new optimizer state and warmup?
>
> Yes, we consider those checkpoints as a new model, with a new optimizer state.
>
> > I assume that you are performing full-parameter fine tuning (no LoRA), both on the final model to add the information, and on the intermediate checkpoint to determine the forget vector?
>
> Yes, we use full fine tuning in all of our experiments.

---

> > ### Author Response · Authors · 2025-11-21
> > **Rebuttal (continued)**
> >
> > > Question 3: It is interesting for me to think about what this method can and cannot do. Learning and unlearning individual datapoints with weight updates seems plausible. But what about more complex behaviors? Have you thought about this?
> >
> > This is a fascinating point. Regarding complex behaviors: identifying the unlearning targets in itself would be a first (challenging) step, requiring a precise understanding of which training examples drive those behaviors. In this work, we focus specifically on machine unlearning, but we view this as complementary to the growing body of work on data attribution, which aims to quantify the influence of individual examples on model predictions and behaviors. These are complementary lines of research: one could imagine that if better methods for data attribution are developed, they would help trace complex behaviors back to specific datapoints or subsets of data, which could then be used as targets for unlearning algorithms such as ours.

---

> > > ### Author Response · Authors · 2025-11-24
> > >
> > > We just realized that we missed this point:
> > >
> > > > The experiments that are ultimately performed are a bit different from what is depicted in Figure 1. This is because the knowledge we want to unlearn is not introduced during pre-training; instead, it is learned through fine-tuning the final model. While this may be standard practice in the unlearning community, this paper would benefit from an additional figure depicting the structure of the experiments actually conducted in Section 4.
> > >
> > > We appreciate the reviewer for raising this detailed point. As also mentioned by the reviewer, it is standard practice in machine unlearning that a model is finetuned on benchmark-related datasets, and a subset of this data is then treated as the unlearning targets. Also, recent work [1] shows that the most challenging unlearning setup is the case where unlearning targets are introduced later in the training. This indicates that current unlearning benchmarks evaluate in this challenging  setting. However, our method (depicted in Figure 1) like other unlearning algorithms is not limited to this setup.
> > > We will make sure to clearly describe the experimental setup of Section 4 and its difference from the general case in the final draft.
> > >
> > > We apologize for missing this point in the initial rebuttal.
> > >
> > > ------------
> > > [1] Yu, Jiatong, et al. "On the Impossibility of Retrain Equivalence in Machine Unlearning." arXiv preprint arXiv:2510.16629 (2025).

---

> > > > ### Comment · Reviewer_bmvB · 2025-11-24
> > > > **Response to Authors**
> > > >
> > > > Thank you for the response. After reading the other reviews and the rebuttal, I maintain my positive assessment of the paper. As a comment, I was most concerned by the missing links to the literature observed by Reviewer eFij, and I think it is critical to clearly outline these connections in the final version, as you promised in the rebuttal.

---

> ### Author Response · Authors · 2025-11-24
>
> We appreciate the reviewer’s positive feedback and for maintaining this great score!
>
> We added an extensive related work section (addressing points raised by reviewer eFij) to Appendix A of the revised draft, and will make sure the connections to these works are clearly articulated in the final version.

---

### Official Review · Reviewer_3L2G · 2025-10-29

**Soundness:** 2
**Presentation:** 2
**Contribution:** 2
**Rating:** 4
**Confidence:** 4

**Summary:**

This paper proposes Model State Arithmetic (MSA) to efficiently eliminate the influence of problematic training data, such as private or copyrighted material. MSA achieves this by leveraging prior model checkpoints, which are model states recorded at different stages of pretraining. Specifically, MSA computes a forget vector by finetuning a checkpoint that precedes the introduction of the unlearning target. This vector captures the data's influence in the weight space and is then arithmetically applied to produce an unlearned model. This design is hypothesized to result in a more precise unlearning compared to previous task arithmetic methods that only use the final trained model.

**Strengths:**

Whereas prior approaches, such as task arithmetic, typically only use the last model parameters, this study uses a model checkpoint that precedes the model's exposure to the unlearning documents. Leveraging these prior model checkpoints yields much more effective unlearning performance, achieving superior or competitive performance compared to prior methods.

MSA consistently achieves superior or competitive performance compared to task vectors across a variety of metrics and scenarios, by using multiple benchmarks and models.

**Weaknesses:**

The rationale for using pre-exposure checkpoints in MSA is intuitive, hypothesizing that it yields more semantically meaningful forget vectors by avoiding entanglement with knowledge already acquired by the final model. However, the study needs stronger justification through empirical analysis (e.g., showing the calculated forget vector is better aligned with the unlearning direction) and a theoretical framework defining the superior properties of vectors derived from early checkpoints versus those derived from fully trained parameters.

The context of Influence Functions (IF) and related literature already encompasses methods designed to estimate the influence of specific training data by considering the timing of their introduction during training, with the goal of approximating the model state where the training data had been absent.
MSA’s mechanism also aims to reproduce a model equivalent to one not trained on the target data by factoring in the training chronology via pre-exposure checkpoints, necessitating a theoritical discussion linking it to prior work.

For instance, the paper "[Data Cleansing for Models Trained with SGD](https://proceedings.neurips.cc/paper_files/paper/2019/hash/5f14615696649541a025d3d0f8e0447f-Abstract.html)" introduces an estimator for SGD-Influence (Equation 2), which shows that the resulting final parameter difference when removing data point $j$: $θ_{-j}^{[T]}-θ^{[T]}$ is approximated by the initial influence of the data point measured at the moment it was learned ($g(z_j ;θ^{[π(j)]})$) multiplied by a sequence of propagation matrices involving ($I−ηH$). Given that MSA calculates the forget vector $g(z_j ;θ^{[π(j)]})$ based on the immediate effect of $z_j$ on a checkpoint $θ^{[\pi(j)]}$, and then applies this vector directly to the final model ($θ_{-j}^{[T]} = θ^{[T]}-g(z_j ;θ^{[π(j)]})$), MSA can arguably be viewed as approximating the complex propagation term ($Z_{T-1}Z_{T-1}, \ldots, Z_{T-1}$ where $Z_t = I-\eta_tH^{[t]}$) with the Identity matrix ($I$).
A necessary theoretical discussion should clarify this connection, explicitly address this approximation ($I−ηH≈I$), and validate its suitability, particularly considering the highly non-convex nature of LLMs.

**Questions:**

How were the checkpoints utilized for MSA (500B, 2207B, 3691B, 3859B trained tokens) selected for OLMo experiments? These values feel somewhat arbitrarily chosen. To eliminate this doubt, it would be better to use checkpoints saved according to a specific rule (for example, using checkpoints saved every 1000B tokens).

---

> ### Author Response · Authors · 2025-11-21
> **Rebuttal**
>
> We appreciate the reviewers observing that using model checkpoints that precede exposure to the unlearning documents—rather than only the last parameters as in task arithmetic—leads to substantially more effective unlearning, and MSA consequently achieves superior or competitive performance compared to task vectors across diverse metrics, benchmarks, and model settings.
>
> > The rationale for using pre-exposure checkpoints in MSA is intuitive, hypothesizing that it yields more semantically meaningful forget vectors by avoiding entanglement with knowledge already acquired by the final model. However, the study needs stronger justification through empirical analysis (e.g., showing the calculated forget vector is better aligned with the unlearning direction) and a theoretical framework defining the superior properties of vectors derived from early checkpoints versus those derived from fully trained parameters.
>
> **Empirical Analysis (e.g.., showing the calculated forget vector is better aligned with the unlearning direction)**
>
> This is demonstrated by downstream performance on unlearning benchmarks. As shown in Tables 1, 2, 3, 4, computing vectors on the final model leads to considerably diminished performance as compared to obtaining update directions from past model checkpoints.
>
> However, we also conduct an additional analysis in the response period to further expand on the reviewer’s suggestion of examining the alignment of the update vectors. Suppose we start from the Llama instruct model and first finetune it on TOFU. We then continue training on C4, and finally train again on TOFU. To apply MSA in this setting, we have five possible checkpoints:
>
> 1. the base model,
> 2. the instruct model,
> 3. the checkpoint after the first TOFU stage,
> 4. the checkpoint after TOFU + C4, and
> 5. the target model trained on TOFU + C4 + TOFU.
>
> The ideal model in this setting is the one trained on TOFU retain authors, then C4, and then TOFU retain authors again.
> The corresponding experimental results are summarized in the following table. As observed there, $MSA_\text{base}$ and $MSA_\text{instruct}$ achieve the best overall performance, while $MSA_\text{TOFU}$ performs worst, followed by $MSA_\text{TOFU+C4}$ and $MSA_\text{TOFU+C4+TOFU}$.
>
> | Model                           | $ACC_\text{forget} \uparrow$ | $ACC_\text{recover} \uparrow$ | $ACC_\text{retain} \uparrow$ | ES on $D_\text{f} \downarrow$ | Model Utility↑| ROUGE$\text{-}L_\text{f} \downarrow$ | Forget Quality$\uparrow$ |
> |---------------------------------|------------------------------|-------------------------------|------------------------------|------------------------------|---------------------------|--------------------------------------|---------------------------|
> | Final (TOFU)                    | 0.04                         | 0.03                          | 0.99                         | 0.94                         | 0.54                      | 0.96                                 | 6.16e-18                  |
> | Ideal (TOFU retain)             | 0.82                         | 0.52                          | 0.99                         | 0.06                         | 0.54                      | 0.38                                 | 0.91                      |
> | $MSA_\text{base}$               | 0.75 (98.7%)                 | 0.37 (96.1%)                  | 0.91 (100.0%)                | 0.08 (100.0%)                | 0.55 (+100%)              | 0.37 (+100%)                          | 0.37                      |
> | $MSA_\text{instruct}$           | 0.76 (100.0%)                | 0.38 (100.0%)                 | 0.89 (98.3%)                 | 0.08 (98.9%)                 | 0.54 (99.9%)              | 0.39 (95.4%)                          | 0.64                      |
> | $MSA_\text{TOFU}$               | 0.67 (88.2%)                 | 0.31 (80.4%)                  | 0.71 (78.5%)                 | 0.09 (80.8%)                 | 0.55 (+100%)              | 0.35 (+100%)                          | 6.86e-10                  |
> | $MSA_{\text{TOFU} + \text{C4}}$ | 0.67 (88.2%)                 | 0.35 (91.5%)                  | 0.89 (98.1%)                 | 0.09 (81.6%)                 | 0.58 (+100%)              | 0.38 (99.6%)                          | 1.83e-05                  |
> | $MSA_{\text{TOFU} + \text{C4} + \text{TOFU}}$ | 0.67 (88.2%)      | 0.30 (79.7%)                  | 0.81 (88.7%)                 | 0.14 (56.4%)                 | 0.54 (99.9%)              | 0.38 (97.5%)                          | 2.77e-09                  |

---

> > ### Author Response · Authors · 2025-11-24
> > **Rebuttal (continued)**
> >
> > When we examine the cosine similarity between each MSA direction and the ideal unlearning direction (ideal model minus final model), we obtain:
> >
> > | variant      | value   |
> > |-------------  |--------:|
> > | base         | 0.095  |
> > | instruct     | 0.132  |
> > | TOFU           | 0.101  |
> > | TOFU+C4     | 0.150  |
> > | TOFU+C4+TOFU  | 0.053  |
> >
> > Interestingly, although \text{MSA}_\text{TOFU+C4} has the highest cosine similarity with the ideal direction, it does not yield the best unlearning performance. This empirical finding suggests that unlearning quality cannot be fully captured by the alignment of the update direction with the ideal one alone, and there is not a clear correlation between behavioral performance and parameter space.
> >
> > **Theoretical Framework**
> >
> > Thank you for the very interesting suggestion. **The contribution type of this work is empirical**.
> > While adding a theoretical framework would be a very interesting exploration and could be a valuable direction for future work, it cannot be considered a major weakness of the existing work.
> > We believe our empirical results establish a strong foundation upon which future theoretical analyses can be built.
> >
> >
> > > How were the checkpoints utilized for MSA (500B, 2207B, 3691B, 3859B trained tokens) selected for OLMo experiments? These values feel somewhat arbitrarily chosen. To eliminate this doubt, it would be better to use checkpoints saved according to a specific rule (for example, using checkpoints saved every 1000B tokens).
> >
> > We note that we took these checkpoints from OLmo-2 checkpoints (https://huggingface.co/allenai/OLMo-2-1124-7B/tree/main).
> > Those checkpoints are stored based on the training steps, however, training steps depend on the specific training recipe for a model e.g., batch size, context length, etc. We instead decided to report the number of tokens that such models are trained on, as a more relevant measure for data-level unlearning.
> >
> > We took the following training steps:
> >
> > 119K steps. 512K steps, 880K steps, 920K steps, with the goal of having the number of tokens between checkpoint and unlearning targets to be around 10B, 100B, 1000B, and more than 3000B tokens away.

---

> > > ### Author Response · Authors · 2025-11-26
> > >
> > > Dear Reviewer 3L2G,
> > >
> > > We’d like to briefly follow up to see if you have any remaining questions or concerns about our rebuttal that we could address before the discussion period closes. Thank you again for your thoughtful review.
> > >
> > > Please let us know if there is anything further we can clarify.

---

### Official Review · Reviewer_qoRY · 2025-10-31

**Soundness:** 2
**Presentation:** 3
**Contribution:** 2
**Rating:** 4
**Confidence:** 4

**Summary:**

This paper focuses on unlearning, i.e., mitigating the influence of specific data points on an already trained model without affecting the whole model. Authors introduce an approach, referred to as Model State Arithmetic, that utilizes model checkpoints saved at different stages of pretraining to estimate and counteract the effect of targeted datapoints. Specifically, a forget vector is computed from a checkpoint that precedes exposure to the unlearning documents and this vector is applied to the final model that has already been trained on the unlearning documents. In addition to the forget vector, a retain vector is also estimated using a small retain set when unavailable and is applied to the final model.

Experiments were conducted using three popular unlearning benchmarks and the proposed approach has been shown to achieve competitive performance when compared to various alternative strategies in the specific setup under consideration (where we have access to an intermediate checkpoint that has not been exposed to the unlearning targets).

**Strengths:**

The proposed approach is simple and easy to use.

While weights arithmetic has been explored before for the problem of unlearning, this paper proposes a simple but effective modification by leveraging intermediate checkpoints.

The paper has been written well and is easy to understand.

**Weaknesses:**

The proposed approach explicitly relies on an intermediate checkpoint that has not been exposed to the unlearning documents. In order to effectively use this weights arithmetic strategy, one needs to have access to the intermediate checkpoints and also know when the unlearning targets were introduced into the training process. Hence, the proposed strategy is applicable only to specific scenarios. For example, one cannot use it for unlearning any given target from an OSS model for which we only have the last checkpoint. Even if we have access to intermediate checkpoints, we need to know exactly when the unlearning target was introduced into the training process so that we can select a checkpoint prior to that. This could be difficult and we may not be able to do it accurately since same facts can appear multiple times in different ways in the pretraining corpus and we may not be able to easily identify them using standard deduplication approaches.


While the paper looks into the effect of number of tokens between the intermediate checkpoint \theta_C and the unlearning targets, it does not study the effect of number of tokens between unlearning targets and the target checkpoint \theta_D. So, it is unclear if the proposed approach would work in situations where we want the model to unlearn something that it has learned a lot of tokens ago.

**Questions:**

What is the effect of the number of tokens between unlearning targets and the final checkpoint \theta_D?

Since access to a checkpoint that is not exposed to unlearning targets is a key element in the proposed approach, authors should discuss how one can effectively identify such checkpoints for any given unlearning target that could appear at multiple locations in the large pertaining corpus. The current experimental setup assumes that we exactly know when the unlearning target is introduced into the training process and in fact assumes that the unlearning target is not too many tokens into the past when compared to checkpoint \theta_D.

---

> ### Author Response · Authors · 2025-11-21
> **Rebuttal**
>
> We appreciate the reviewer recognizing the simplicity, practicality, and effectiveness of our approach. We are also glad that the reviewer found the paper well written and easy to understand.
>
>
> > one needs to have access to intermediate checkpoints, and also know when unlearning targets are introduced in the training process
>
> We discuss precisely this in Lines 197-204 and Lines 205-215. Briefly,
>
> + **Access to intermediate checkpoints (Lines 197-204)**.
> As the reviewer notes, the MSA method is only applicable to specific scenarios (ones where intermediate checkpoints are available), but we believe this includes practically and legally significant deployment scenarios— such as enabling model providers to support the RTBF (the right to be forgotten from General Data Protection Regulation), where regulation would require model providers to delete particular data instances from the model upon request from a data subject, before releasing the model to the public. Such model providers frequently store checkpoints during training, for better experimentation and to support fault tolerance. However, MSA can also be implemented for local versions of open models that publicly release checkpoints, such as models from the (widely-used) OLMo and Pythia families.
>
> + **Access to effective checkpoints preceding the unlearning target (Lines 205-215)**
> While we have found that obtaining a checkpoint before the unlearning target is helpful, this need not be a very precise localization process— we find that checkpoints that precede the target by even hundreds of billions of tokens can be useful (Lines 421-460). Moreover, we also note that training recipes and the data used are usually known to the provider in advance, and we hope that just as providers have found that maintaining indexes of training data [3, 4] has a broad range of uses, such as shedding light on questions about attribution [5, 6]) and contamination ([3], practitioners also invest in maintaining indexes of when models encounter information during training, due to the utility of techniques like MSA which can make use of model state history, and to support efforts in studying how language models store, learn, and update knowledge. Indeed, we are already starting to witness research[1] developing techniques for indexing related information during training.
>
>
> > While the paper looks into the effect of the number of tokens between the intermediate checkpoint \theta_C and the unlearning targets, it does not study the effect of number of tokens between unlearning targets and the target checkpoint \theta_D. So, it is unclear if the proposed approach would work in situations where we want the model to unlearn something that it has learned a lot of tokens ago.
>
> This is an important point, and we appreciate that the reviewer highlighted this. We address this below:
>
> First, we would like to note that existing machine unlearning benchmarks typically assume that the unlearning targets are introduced at the end of training, and we largely follow this setup to enable fair comparison with prior unlearning algorithms. Recent work such as [2] has begun to study how the position of the unlearning targets in the training trajectory affects unlearning performance, and shows that the most challenging setting is indeed when the targets are introduced late in training. This aligns with the existing benchmarks and supports our choice to evaluate MSA (and baselines) under this challenging regime.
> That said, we would like to discuss the scenarios the reviewer highlighted: that is, investigating the effect of number of tokens between unlearning targets and the target checkpoint $\theta_D$.
>
> To investigate this, during the response period we conducted an additional experiment. We first finetune Llama-3.2-1B-instruct on TOFU, and then further finetune it on $\sim 20$M tokens of C4. The ideal model (which has not been exposed to the unlearning target)  in this setup is the one trained on the “retain” subset of TOFU and further finetuned on C4. The following table (Table 10 in revised draft in Appendix F) reports the results in this scenario.
> As seen there, MSA variants that use checkpoints taken before the introduction of the unlearning targets ($MSA_\text{base}$ and $MSA_\text{instruct}$) remain effective and achieve values close to the ideal model, even though the unlearning targets now lie many tokens before the final checkpoint. In contrast, using a checkpoint after seeing the unlearning targets ($MSA_\text{TOFU}$), but before the model encounters the C4 tokens,  underperforms on multiple metrics.

---

> ### Author Response · Authors · 2025-11-21
> **Rebuttal (contined)**
>
> | Model            | $ACC_\text{forget}$↑   | $ACC_\text{recover}$↑ | $ACC_\text{retain}$↑ | ES on $D_\text{f}$↓   | Model Utility↑     | ROUGE-$L_\text{f}$↓  | Forget Quality↑ |
> |------------------|-----------------------|----------------------|---------------------|----------------------|----------------------|---------------------|------------------|
> | Final (TOFU)     | 0.48                  | 0.24                 | 0.66                | 0.19                 | 0.55                 | 0.49                | 9.34e-13         |
> | Ideal (TOFU retain) | 0.83               | 0.45                 | 0.69                | 0.07                 | 0.55                 | 0.38                | 1.31e-04         |
> | $MSA\_\text{base}$    | 0.79 (95.5%)       | 0.39 (87.6%)         | 0.68 (98.2%)        | 0.06 (+100%)         | 0.53 (97.8%)         | 0.34 (+100%)        | 0.42            |
> | $MSA\_\text{instruct}$ | 0.83 (100.0%)      | 0.45 (+100%)         | 0.70 (+100%)        | 0.06 (+100%)         | 0.55 (+100%)         | 0.36 (+100%)        | 0.70            |
> | $MSA\_\text{TOFU}$     | 0.73 (88.2%)       | 0.37 (82.6%)         | 0.70 (+100%)        | 0.08 (80.2%)         | 0.57 (+100%)         | 0.33 (+100%)        | 1.10e-09        |
>
> These results provide empirical evidence that MSA can still work well when the model is asked to forget information learned a significant number of tokens earlier, while reinforcing our earlier observation that checkpoints taken after exposure to the forget set are less suitable for constructing effective unlearning updates. We have added this analysis to Appendix F.
>
> > Since access to a checkpoint that is not exposed to unlearning targets is a key element in the proposed approach, authors should discuss how one can effectively identify such checkpoints for any given unlearning target that could appear at multiple locations in the large pertaining corpus.
>
> We appreciate the reviewer for raising this very interesting consideration.
> During the rebuttal we conducted an additional experiment that simulates a setting where the forget data appears multiple times and is not always close to the final checkpoint. We take Llama-3.2-1B-instruct, first finetune it on TOFU, then train it on a subset of C4 (~20M tokens), and finally finetune again on TOFU. This final model (TOFU + C4 + TOFU) is the target of unlearning. The ideal model in this setup is trained on TOFU retain, then C4, then TOFU retain again. The following table (Table 11 in Appendix G of the revised draft) reports the empirical results.
>
> In this configuration, we have five natural checkpoints at which to apply MSA: (1) the base model, (2) the instruct model, (3) the model after the first TOFU stage, (4) the model after TOFU + C4, and (5) the final model after TOFU + C4 + TOFU. As seen in Table, when MSA leverages checkpoints that precede any exposure to TOFU ($MSA_\text{base}$, $MSA_\text{instruct}$), it achieves strong performance, with values close to the ideal model. In contrast, using checkpoints that have already seen TOFU systematically underperforms. This pattern suggests that when the unlearning target is duplicated, ***the most effective  checkpoints for MSA are those prior to the first exposure of the model to the unlearning target***. We have added this analysis to Appendix G.

---

> > ### Author Response · Authors · 2025-11-21
> > **Rebuttal (continued)**
> >
> > Indeed, this work represents an important first step toward leveraging model checkpoints for LLM unlearning under realistic constraints. A comprehensive investigation of how to unlearn duplicated data—ranging from highly frequent pre-training examples to data duplicated only a few times or even seen just once— offers a promising direction for future work.
> >
> > | Model                           | $ACC_\text{forget} \uparrow$ | $ACC_\text{recover} \uparrow$ | $ACC_\text{retain} \uparrow$ | ES on $D_\text{f} \downarrow$ | Model Utility↑| ROUGE$\text{-}L_\text{f} \downarrow$ | Forget Quality$\uparrow$ |
> > |---------------------------------|------------------------------|-------------------------------|------------------------------|------------------------------|---------------------------|--------------------------------------|---------------------------|
> > | Final (TOFU)                    | 0.04                         | 0.03                          | 0.99                         | 0.94                         | 0.54                      | 0.96                                 | 6.16e-18                  |
> > | Ideal (TOFU retain)             | 0.82                         | 0.52                          | 0.99                         | 0.06                         | 0.54                      | 0.38                                 | 0.91                      |
> > | $MSA_\text{base}$               | 0.75 (98.7%)                 | 0.37 (96.1%)                  | 0.91 (100.0%)                | 0.08 (100.0%)                | 0.55 (+100%)              | 0.37 (+100%)                          | 0.37                      |
> > | $MSA_\text{instruct}$           | 0.76 (100.0%)                | 0.38 (100.0%)                 | 0.89 (98.3%)                 | 0.08 (98.9%)                 | 0.54 (99.9%)              | 0.39 (95.4%)                          | 0.64                      |
> > | $MSA_\text{TOFU}$               | 0.67 (88.2%)                 | 0.31 (80.4%)                  | 0.71 (78.5%)                 | 0.09 (80.8%)                 | 0.55 (+100%)              | 0.35 (+100%)                          | 6.86e-10                  |
> > | $MSA_{\text{TOFU} + \text{C4}}$ | 0.67 (88.2%)                 | 0.35 (91.5%)                  | 0.89 (98.1%)                 | 0.09 (81.6%)                 | 0.58 (+100%)              | 0.38 (99.6%)                          | 1.83e-05                  |
> > | $MSA_{\text{TOFU} + \text{C4} + \text{TOFU}}$ | 0.67 (88.2%)      | 0.30 (79.7%)                  | 0.81 (88.7%)                 | 0.14 (56.4%)                 | 0.54 (99.9%)              | 0.38 (97.5%)                          | 2.77e-09                  |
> >
> >
> > ---------------
> > ##### [1] Gottesman, Daniela, et al. "LMEnt: A Suite for Analyzing Knowledge in Language Models from Pretraining Data to Representations." arXiv preprint arXiv:2509.03405 (2025).
> > ##### [2] Yu, Jiatong, et al. "On the Impossibility of Retrain Equivalence in Machine Unlearning." arXiv preprint arXiv:2510.16629 (2025).
> > ##### [3] Elazar, Yanai, et al. "What's In My Big Data?." arXiv preprint arXiv:2310.20707 (2023).
> > ##### [4] Liu, Jiacheng, et al. "Infini-gram: Scaling unbounded n-gram language models to a trillion tokens." arXiv preprint arXiv:2401.17377 (2024).
> > ##### [5] Liu, Jiacheng, et al. "OLMoTrace: Tracing language model outputs back to trillions of training tokens." Proceedings of the 63rd Annual Meeting of the Association for Computational Linguistics (Volume 3: System Demonstrations). 2025.
> > ##### [6] Ravichander, Abhilasha, et al. "Halogen: Fantastic llm hallucinations and where to find them." arXiv preprint arXiv:2501.08292 (2025).

---

> > > ### Author Response · Authors · 2025-11-26
> > >
> > > Dear Reviewer qoRY, we wanted to follow up to see if you have any additional questions or concerns about our response that we could address before the discussion period closes.
> > >
> > > Thank you again for taking the time to review our paper. Please don't hesitate to let us know if there is anything else you would like us to clarify.

---

### Official Review · Reviewer_eFij · 2025-11-05

**Soundness:** 3
**Presentation:** 2
**Contribution:** 2
**Rating:** 4
**Confidence:** 4

**Summary:**

The paper hypothesizes that using earlier checkpoints (specifically those before seeing the forget data) in training to decide the unlearning update applied to the final model can outperform common methods which only see the final model. They devise a specific method which compute an unlearning update by training a checkpoint on the forget set and retrain set seperately and adding a weighted sum of the directions. Comparison against past methods on several benchmarks show the method matches or outperforms past methods. In particular, for ToFU they introduce several new metrics to test additionally to the previous metrics, and show they help distinguish methods (e.g., the failures of RMU) better.

**Strengths:**

1) The experiments are thorough, including several benchmarks and additional metrics over past work

2) The experiments generally support that the method performs better than the tested baselines

**Weaknesses:**

I mention several concerns below; I am happy to reconsider my score given the authors can resolve some of these. See the questions for a breakdown of the concerns into specific questions.

1) Contrary to the paper’s claim, this is not the first work to use previous model states to compute the unlearning. In fact some of the first approximate unlearning methods used previous model checkpoints to compute the model update: Graves et al., [1] compute the update by using several checkpoints during training, while Thudi et al., study how effectively one can unlearn by computing the unlearning update from the model checkpoint before training on the dataset containing the forget data [2]. More recent work has even investigated how to analyze the guarantees of unlearning by restarting training from an earlier checkpoint [3]. No mention or comparison to this existing literature is made.

2) A claim of this paper is that computing updates with checkpoints before seeing the forget data is better than using models that had already trained on the data. However, there are no experiments ablating performance as the proposed method is applied to checkpoints obtained at different stages of training with the forget data; the authors instead compare to other methods. The current evaluation hence doesn’t seems to answer whether the method actually does better at checkpoints before seeing the forget data than when applied to checkpoints after; in fact the method gets better as it is applied to models with more training.

3) Moreover, if the claim is to understand the importance of checkpoints, it seems reasonable to also ablate the other methods by applying them to earlier checkpoints and understand how important their specific unlearning update direction is. This relates back to point 1 where methods using previous checkpoints already exist and comparing to other methods would help disentangle the role of the checkpoint to the proposed method.

[1] Graves, Laura, Vineel Nagisetty, and Vijay Ganesh. "Amnesiac machine learning." Proceedings of the AAAI Conference on Artificial Intelligence. Vol. 35. No. 13. 2021.

[2] Thudi, Anvith, et al. "Unrolling sgd: Understanding factors influencing machine unlearning." 2022 IEEE 7th European Symposium on Security and Privacy (EuroS&P). IEEE, 2022.

[3] Mu, Siqiao, and Diego Klabjan. "Rewind-to-delete: Certified machine unlearning for nonconvex functions." arXiv preprint arXiv:2409.09778 (2024).

**Questions:**

Given my previously mentioned concerns, I have the following questions which can answer them.

1) Could the authors tone down the claims of the paper to focus more on the impact of the checkpoint used to compute the unlearning update than using checkpoints altogether; as mentioned past work has already proposed methods that use certain checkpoints, but this paper can add to this literature by focusing itself on empirically investigating which checkpoints lead to better unlearning.

2) On the above, could the authors clarify how their specific method works when applied to checkpoints obtained when training on the forget data; are their implicit findings somewhere in the paper and I somehow missed them?

3) Specifically can the authors claim their method applied to a model fully trained on the forget data performs worse than their method applied to the pre-trained checkpoint (before seeing the forget data)?

4) Furthermore, can the authors have evidence for why the best checkpoint to use is the one just before training on the forget data and not one that comes after starting to train on the forget data?

5) Do the authors have results on what happens when other methods are applied on earlier checkpoints? E.g., one can think of rewind-to-delete as applying fine-tuning on the retain set at an earlier checkpoint, and one could do the same with other methods.

---

> ### Author Response · Authors · 2025-11-21
> **Rebuttal**
>
> We appreciate the reviewer’s feedback. We are glad they found our experimental setup thorough, recognizing that we evaluate on several benchmarks and additional metrics compared to work in the area, and that our results support the effectiveness and competitiveness of our method. We also appreciate that the reviewer is willing to raise their score, based on resolving some of their concerns. We address each of these points in detail below and remain fully available to clarify any remaining question before the deadline.
>
> ### Weaknesses
>
> > Contrary to the paper’s claim, this is not the first work to use previous model states to compute the unlearning [1, 2, 3].
>
> The proposed method is the first *post-hoc* unlearning method that uses *intermediate model checkpoints* to achieve strong empirical effectiveness at unlearning data from *large language models*. We thank the reviewer for bringing the additional related work to our attention. Below, we describe how they are related and compare our proposed method with them:
>
> **Amnesiac Machine Unlearning [1].** Although conceptually related, since it also exploits information from the model’s training trajectory, this method faces two key limitations that make it impractical for large language models:
> + First, it requires logging and storing the full parameter update vector for every training step whose batch might later be subject to deletion, along with a record of which examples appear in which batches. In realistic deletion scenarios, this implies maintaining an $O(steps \times \| \theta \|)$ log of updates, which is vastly larger than the handful of checkpoints typically retained in LLM training and becomes prohibitive at the scale large language models are trained at (often multi-billion-parameter models trained on trillions of tokens). *To our knowledge, amnesiac unlearning has never been implemented for large language models* (and it is unclear if it is feasible).
> + Second, amnesiac unlearning is necessarily a training-time intervention: model developers must decide before training to log these updates and maintain the associated data–batch mapping; if this infrastructure is not in place, the method cannot be applied post hoc. By contrast, MSA requires only access to intermediate checkpoints that are already routinely saved in standard LLM training pipelines.
>
> Combined, these make MSA more practical for large language models and enable post-hoc unlearning, as demonstrated by our application to existing models such as OLMo, without any prior modifications or special preparation during training.
>
>
> **Unrolling SGD [2].** Thank you for bringing this work to our attention— this method is conceptually very similar and we will clearly position our work with respect to this method as described below.
>
> Summary of work: The authors study approximate machine unlearning by analyzing SGD and proposing verification error, defined as the distance in weight space between an approximately unlearned model and the ideal retrained model. They introduce (i) single-gradient unlearning, which given an initial model state $w_{0}$, reverses the gradient of the forget sample with respect to that model state to approximate removal , and (ii) a training-time regularizer that constrains the SGD trajectory to make future unlearning requests easier. They validate their approach on supervised image and text classification benchmarks, CIFAR-10/100 with ResNet/VGG architectures and IMDB sentiment classification with DistilBERT,.
>
> Clear claim position: We will clearly position our work with respect to unrolling SGD as follows. The concept of single-gradient unlearning is conceptually similar, as it also leverages information about the forget set from a past model state to perform approximate unlearning. We will clearly describe that *we are not the first to look at using a previous model state to compute gradients for forgetting*, and point to the unrolling SGD work which uses vectors derived from a pretrained model state (similar to $MSA_{\text{base}}$), and an initial model state.

---

> ### Author Response · Authors · 2025-11-21
> **Rebuttal (continued)**
>
> **Unrolling SGD [2] (continued).**
> However, our approach differs in the following aspects: (1) First, our method is fully post-hoc and does not require any intervention in the original training objective or optimizer for empirical effectiveness (applying regularization during training was found to be very important for effective unlearning in the unrolling SGD method) (2) Second, we demonstrate the empirical effectiveness of MSA using a comprehensive suite of benchmarks and metrics, including recent unlearning benchmarks and behavior-level measures, rather than focusing primarily on verification or unlearning error in parameter space. (3) Third, we empirically demonstrate that MSA is effective at LLM scale, with large models trained on billions of tokens. In contrast to the experimental setup of~[2], which assumes access to a model checkpoint taken immediately before the introduction of the unlearning targets, we conduct real-scale experiments using checkpoints that may lie billions of tokens before the forget set. (4) We empirically investigate intermediate model checkpoints rather than only an initial random-weight initialization or a base model to compute model updates, and (5) Finally, the empirical performance reported in [2] appears to sharply degrade when the training-time regularization term is removed, whereas our method achieves strong empirical performance in a purely post-hoc setting without any modification to the original training process.
>
> **Rewind-to-Delete [3].** This method falls outside the common efficiency criteria for approximate machine unlearning, where the unlearning cost is expected to scale with the size of the forget set rather than the retain set. Rewind-to-Delete leverages an earlier checkpoint and retrains it on the retain set, achieving valuable certified guarantees, but its cost scales with the size of the retained data. Consequently, it does not fit within the typical efficiency regime of approximate unlearning methods for large language models. whose complexity is $O(|D_\text{f}|)$, such as MSA, NPO, and GradDiff in the LLM setting.
>
> While these works [1, 2, 3] are conceptually related and we acknowledge them in the extended related work section, they operate under different assumptions (e.g., training-time interventions, specific optimizer choices, or retain-set scaled retraining with certified guarantees) than our setting (which is intended to work at the scale of modern large language models). For these reasons, we do not include a direct empirical comparison in our draft.
>
> **We added this explicit claim to Appendix A of the revised draft**:
>
> "we propose MSA as an efficient approximate unlearning algorithm whose runtime scales as $O(|D_\text{f}|)$, similar to other efficient approximate unlearning methods, while explicitly leveraging model checkpoints under the constraints of LLM training pipelines. Unlike prior approaches [1, 2], MSA does not require storing training parameter updates or having control over the training objective or optimizer, and instead operates purely post hoc on existing checkpoints. Across multiple benchmarks and evaluation metrics, MSA achieves competitive, and often superior, performance compared to prior baselines. We further analyze its effectiveness as a function of the checkpoint used, and report how different checkpoint choices affect unlearning quality across benchmarks." and that "we are not the first to look at previous checkpoints to compute gradients to forget".
>
> > A claim of this paper is that computing updates with checkpoints before seeing the forget data is better than using models that had already trained on the data. However, there are no experiments ablating performance as the proposed method is applied to checkpoints obtained at different stages of training with the forget data; the authors instead compare to other methods. The current evaluation hence doesn’t seems to answer whether the method actually does better at checkpoints before seeing the forget data than when applied to checkpoints after; in fact the method gets better as it is applied to models with more training.
>
> This comparison is present. In our setting, applying MSA directly to the target model (i.e., the model already trained on the unlearning targets) reduces to the Task Vector method [4], which obtains the forget direction from a model that has already seen the forget data. We compare MSA against Task Vector across all baselines, and as shown in our experiments, MSA consistently outperforms Task Vector on several metrics. For a more detailed explanation of this connection, we refer the reviewer to the last paragraph of Section 3 in the draft (Lines 235-240)

---

> > ### Author Response · Authors · 2025-11-21
> > **Rebuttal (continued)**
> >
> > > Moreover, if the claim is to understand the importance of checkpoints, it seems reasonable to also ablate the other methods by applying them to earlier checkpoints and understand how important their specific unlearning update direction is. This relates back to point 1 where methods using previous checkpoints already exist and comparing to other methods would help disentangle the role of the checkpoint to the proposed method.
> >
> > We note that our work introduces the idea of leveraging intermediate model checkpoints in the LLM regime to achieve efficient yet competitive unlearning algorithms. We agree with the reviewer that there are multiple ways to exploit such checkpoints, and in this paper we propose using them to obtain both forget and retain directions for unlearning. We believe this perspective can open future research directions on how to more effectively leverage intermediate checkpoints for unlearning.
> >
> > To shed more light on the reviewer’s suggestion, we implement their proposed idea of applying other unlearning techniques to intermediate checkpoints, and then using the resulting update direction. More specifically, let $\theta_0$ be an intermediate checkpoint. We apply a baseline unlearning algorithm starting from $\theta_0$, obtaining a model $\theta_1$. We then extract the change direction $\theta_1 - \theta_0$ and apply it to the target model $\theta_{\mathcal{D}}$ with a tunable scalar $\alpha$, yielding
> >
> > $$\theta_{\text{unlearn}} = \theta_{\mathcal{D}} + \alpha (\theta_1 - \theta_0).$$
> >
> > We select the optimal value of $\alpha$ via validation search, as we do for other methods.
> >
> > The following new table (Table 12 in Appendix H of the revised draft) reports experimental results on the TOFU (forget10) task with Llama-3.2-1B, where other unlearning algorithms are augmented with model checkpoints following the above procedure. For example, when applying NPO, we denote $NPO_{\text{base}}$ and $NPO_{\text{instruct}}$ for NPO applied to the pretrained base model and the instruct model, respectively, while NPO alone refers to the case where it is applied to the target model.
> >
> > As seen in this table, these algorithms do not benefit from leveraging intermediate checkpoints in this way; they are outperformed by our method and typically exhibit degraded performance compared to their standard variants applied directly to the unlearning targets. We have added this analysis to Appendix G.

---

> ### Author Response · Authors · 2025-11-21
> **Rebuttal (continued)**
>
> | Model                          | $ACC_\text{forget} \uparrow$ | $ACC_\text{recover} \uparrow$ | $ACC_\text{retain} \uparrow$ | ES on $D_\text{f} \downarrow$ | Model Utility $\uparrow$ | ROUGE$\text{-}L_\text{f} \downarrow$ | Forget Quality $\uparrow$ |
> |--------------------------------|------------------------------|-------------------------------|------------------------------|------------------------------|---------------------------|--------------------------------------|---------------------------|
> | Final (TOFU)                   | 0.05                         | 0.03                          | 0.98                         | 0.87                         | 0.52                      | 0.94                                 | 1.12e-19                  |
> | Ideal (TOFU retain)            | 0.82                         | 0.98                          | 0.98                         | 0.06                         | 0.51                      | 0.38                                 | 1.0                       |
> | $MSA_\text{base}$              | 0.79 (96.6%)                 | 0.39 (89.1%)                  | 0.87 (89.2%)                 | 0.06 (+100%)                 | 0.55 (+100%)              | 0.32 (+100%)                          | 0.02                      |
> | $MSA_\text{instruct}$          | 0.81 (99.1%)                 | 0.44 (100.0%)                 | 0.85 (87.1%)                 | 0.06 (+100%)                 | 0.52 (+100%)              | 0.37 (+100%)                          | 0.28                      |
> | NPO                            | 0.66 (81.0%)                 | 0.25 (57.7%)                  | 0.92 (94.1%)                 | 0.12 (50.4%)                 | 0.54 (+100%)              | 0.31 (+100%)                          | 3.25e-04                  |
> | NPO (base)                     | 0.76 (92.4%)                 | 0.29 (66.9%)                  | 0.53 (54.5%)                 | 0.06 (+100%)                 | 0.27 (52.5%)              | 0.24 (+100%)                          | 9.99e-07                  |
> | NPO (instruct)                 | 0.67 (81.3%)                 | 0.24 (54.3%)                  | 0.71 (72.8%)                 | 0.11 (58.3%)                 | 0.50 (96.6%)              | 0.27 (+100%)                          | 1.02e-13                  |
> | RMU                            | 0.85 (+100%)                 | 0.10 (22.9%)                  | 0.97 (100.0%)                | 0.06 (+100%)                 | 0.52 (+100%)              | 0.25 (+100%)                          | 0.94                      |
> | RMU (base)                     | 0.95 (+100%)                 | 0.04 (8.6%)                   | 0.36 (37.0%)                 | 0.04 (+100%)                 | 0.35 (68.5%)              | 0.20 (+100%)                          | 5.00e-05                  |
> | RMU (instruct)                 | 0.77 (93.6%)                 | 0.19 (43.4%)                  | 0.77 (78.7%)                 | 0.08 (81.9%)                 | 0.48 (92.7%)              | 0.32 (+100%)                          | 1.49e-16                  |
> | GradDiff                       | 0.46 (56.6%)                 | 0.21 (48.6%)                  | 0.90 (92.0%)                 | 0.22 (28.4%)                 | 0.54 (+100%)              | 0.42 (88.8%)                          | 6.03e-11                  |
> | GradDiff (base)                | 0.60 (74.0%)                 | 0.20 (45.1%)                  | 0.61 (62.7%)                 | 0.09 (67.3%)                 | 0.41 (80.8%)              | 0.38 (98.3%)                          | 6.16e-18                  |
> | GradDiff (instruct)            | 0.75 (91.7%)                 | 0.15 (34.3%)                  | 0.40 (41.1%)                 | 0.08 (77.4%)                 | 0.22 (42.2%)              | 0.29 (+100%)                          | 5.63e-20                  |
> | SatImp                         | 0.72 (87.8%)                 | 0.28 (63.4%)                  | 0.77 (78.9%)                 | 0.07 (93.8%)                 | 0.51 (+100%)              | 0.31 (+100%)                          | 1.30e-05                  |
> | SatImp (base)                  | 0.82 (+100%)                 | 0.15 (34.3%)                  | 0.31 (31.6%)                 | 0.05 (+100%)                 | 0.25 (49.3%)              | 0.30 (+100%)                          | 1.07e-08                  |
> | SatImp (instruct)              | 0.72 (88.1%)                 | 0.21 (47.4%)                  | 0.51 (51.9%)                 | 0.07 (94.0%)                 | 0.28 (54.7%)              | 0.30 (+100%)                          | 2.24e-17                  |

---

> ### Author Response · Authors · 2025-11-21
> **Rebuttal (continued)**
>
> | Model                          | $ACC_\text{forget} \uparrow$ | $ACC_\text{recover} \uparrow$ | $ACC_\text{retain} \uparrow$ | ES on $D_\text{f} \downarrow$ | Model Utility $\uparrow$ | ROUGE$\text{-}L_\text{f} \downarrow$ | Forget Quality $\uparrow$ |
> |--------------------------------|------------------------------|-------------------------------|------------------------------|------------------------------|---------------------------|--------------------------------------|---------------------------|
> | Final (TOFU)                   | 0.05                         | 0.03                          | 0.98                         | 0.87                         | 0.52                      | 0.94                                 | 1.12e-19                  |
> | Ideal (TOFU retain)            | 0.82                         | 0.98                          | 0.98                         | 0.06                         | 0.51                      | 0.38                                 | 1.0                       |
> | $MSA_\text{base}$              | 0.79 (96.6%)                 | 0.39 (89.1%)                  | 0.87 (89.2%)                 | 0.06 (+100%)                 | 0.55 (+100%)              | 0.32 (+100%)                          | 0.02                      |
> | $MSA_\text{instruct}$          | 0.81 (99.1%)                 | 0.44 (100.0%)                 | 0.85 (87.1%)                 | 0.06 (+100%)                 | 0.52 (+100%)              | 0.37 (+100%)                          | 0.28                      |
> | UNDIAL                         | 0.52 (63.9%)                 | 0.26 (58.3%)                  | 0.89 (91.0%)                 | 0.04 (+100%)                 | 0.54 (+100%)              | 0.31 (+100%)                          | 7.98e-17                  |
> | UNDIAL (base)                  | 0.78 (95.1%)                 | 0.11 (24.6%)                  | 0.39 (40.4%)                 | 0.06 (+100%)                 | 0.40 (77.8%)              | 0.29 (+100%)                          | 1.49e-16                  |
> | UNDIAL (instruct)              | 0.82 (+100%)                 | 0.10 (22.3%)                  | 0.39 (39.8%)                 | 0.06 (+100%)                 | 0.41 (79.8%)              | 0.23 (+100%)                          | 1.12e-19                  |

---

> > ### Author Response · Authors · 2025-11-21
> > **Rebuttal (continued)**
> >
> > ### Questions
> >
> > > Could the authors tone down the claims of the paper to focus more on the impact of the checkpoint used to compute the unlearning update than using checkpoints altogether; as mentioned past work has already proposed methods that use certain checkpoints, but this paper can add to this literature by focusing itself on empirically investigating which checkpoints lead to better unlearning.
> >
> > Claim positioning: We do not, and will not, claim to be the first unlearning method to use a past model state. Additionally we will use the following language in the draft to make the claim clearer “ the proposed method is the first *post-hoc* unlearning method that uses *intermediate model checkpoints* to achieve strong empirical performance at unlearning data from *large language models*.” Our central claim is one of empirical efficacy: we demonstrate that this procedure outperforms existing unlearning algorithms on several important metrics, at the scale of modern language models.
> > This is added to Appedix of A of revised draft.
> >
> > > On the above, could the authors clarify how their specific method works when applied to checkpoints obtained when training on the forget data; are their implicit findings somewhere in the paper and I somehow missed them?
> >
> > Addressed in response to W2.
> >
> > > Specifically can the authors claim their method applied to a model fully trained on the forget data performs worse than their method applied to the pre-trained checkpoint (before seeing the forget data)?
> >
> > Yes, we denote the results of unlearning by taking the checkpoint after introduction of unlearning targets as Task Vectors in our Tables across all benchmarks  [4].
> >
> > > Furthermore, can the authors have evidence for why the best checkpoint to use is the one just before training on the forget data and not one that comes after starting to train on the forget data?
> >
> > As shown in our experimental results, when a checkpoint is taken after the introduction of the unlearning targets into the training pipeline (Task Vector), unlearning performance degrades. In contrast, leveraging checkpoints prior to exposure to the unlearning targets typically yields competitive or superior performance. In our experiments, the closer these prior checkpoints are to the point where the forget data is introduced, the better the unlearning quality tends to be (see Lines 421–460) That said, using checkpoints even hundreds of billions of tokens earlier still leads to competitive performance that outperforms other unlearning baselines and remains close to MSA with later (closer) checkpoints.
> >
> > We also observe that the sensitivity to checkpoint distance is benchmark-dependent: for example, on RESTOR [5], the drop in performance when using substantially earlier checkpoints (trillions of tokens before) is negligible. Please see lines [420–450] for further details.
> >
> > > Do the authors have results on what happens when other methods are applied on earlier checkpoints? E.g., one can think of rewind-to-delete as applying fine-tuning on the retain set at an earlier checkpoint, and one could do the same with other methods.
> >
> > Addressed in “response to W3”.
> >
> > -----------------
> >
> > #### [1] Graves, Laura, Vineel Nagisetty, and Vijay Ganesh. "Amnesiac machine learning." Proceedings of the AAAI Conference on Artificial Intelligence. Vol. 35. No. 13. 2021.
> > ####  [2] Thudi, Anvith, et al. "Unrolling sgd: Understanding factors influencing machine unlearning." 2022 IEEE 7th European Symposium on Security and Privacy (EuroS&P). IEEE, 2022.
> > #### [3] Mu, Siqiao, and Diego Klabjan. "Rewind-to-delete: Certified machine unlearning for nonconvex functions." arXiv preprint arXiv:2409.09778 (2024).
> > #### [4] Ilharco, Gabriel, et al. "Editing models with task arithmetic." arXiv preprint arXiv:2212.04089 (2022).
> > #### [5] Rezaei, Keivan, et al. "RESTOR: Knowledge Recovery in Machine Unlearning." arXiv preprint arXiv:2411.00204 (2024).

---

> > > ### Comment · Reviewer_eFij · 2025-11-24
> > > **Response to Authors**
> > >
> > > I appreciate the author's response and have raised my score.
> > >
> > > A few minor questions and suggestions:
> > >
> > > 1) As a suggestion for the final draft, I recommend perhaps plotting the attached table to make the trends more apparent.
> > >
> > > 2) Make sure to clearly define post-hoc unlearning methods; technically Amnesiac unlearning can be applied post-hoc if given a few checkpoints (just re-compute the gradient step). In the end I think the paper makes an interesting observation about task vector methods and can position itself to be focused on task vector like methods.

---

> > > > ### Author Response · Authors · 2025-11-24
> > > >
> > > > We are very grateful for the increased score!
> > > >
> > > >
> > > >
> > > > > As a suggestion for the final draft, I recommend perhaps plotting the attached table to make the trends more apparent
> > > >
> > > > We appreciate the reviewer's feedback. We will include a (likely bar) plot in the final version to provide a clearer visualization of the table.
> > > >
> > > > > Make sure to clearly define post-hoc unlearning methods; technically Amnesiac unlearning can be applied post-hoc if given a few checkpoints (just re-compute the gradient step).
> > > >
> > > > Thanks for raising this point. We will more clearly define post-hoc unlearning methods and explicitly clarify how our method relates to Amnesiac unlearning.
> > > >
> > > > > In the end I think the paper makes an interesting observation about task vector methods and can position itself to be focused on task vector like methods.
> > > >
> > > > We appreciate the reviewer’s positive feedback on our observations!

---

### Author Response · Authors · 2025-11-21
**Revised draft**

Based on the reviewers’ valuable feedback, we have revised the draft as follows. These are the changes made in this revision:

- **Appendix A**: Extended related work (reviewer *eFij*).
- **Appendix F**: Unlearning targets introduced many tokens before the final checkpoint (reviewer *qoRY*).
- **Appendix G**: Unlearning with repeated exposure to TOFU (reviewer *qoRY*).
- **Appendix H**: Augmenting baselines with intermediate checkpoints (reviewer *eFij*).
- **Appendix I**: Potential overlap with pretraining data (reviewer *bmvB*).
- **Figure 1**: Edited as requested by reviewer *bmvB*.

The revised draft has been uploaded.

---

### Author Response · Authors · 2025-12-03
**Note to AC**

Dear AC,

We appreciate your time dedicated to this paper. In light of the reset, we would like to ensure you have the context of the discussion period before it closed. Prior to the reset, our paper had achieved a consensus for strong acceptance from the engaged reviewers.
Key updates from the discussion period include:


+ **Reviewer eFij (Engaged, Score improved 4 → 8):** Following our detailed response, this reviewer fully supported acceptance.
+ **Reviewer bmv8 (Engaged, Score maintained at 8):** Maintained strong support after reviewing the rebuttal and other reviews.
+ **Reviewer 3L2G (Did not engage):** We addressed their concerns about whether the forget vector computed from earlier checkpoints is better aligned with the true unlearning direction in two ways: (1) First, we pointed them to the baseline already included in the draft, where the forget vector is computed from a checkpoint after the model has been exposed to the unlearning target (the “task vector” baseline). This baseline consistently underperforms compared to using a vector derived from a checkpoint before exposure, demonstrating that the pre-exposure forget vector is in fact more effective, and (2)  Second, we provided empirical evidence showing  that simple parameter space analysis is often a weak proxy for model behavior. Finally, we emphasized that while a formal theoretical analysis of alignment between forget vectors and unlearning directions would indeed be valuable, it is beyond the scope of this empirical study. We highlighted it as a promising direction for future work.
+ **Reviewer qoRY (Did not engage)**: We addressed their concerns regarding practical considerations around using model checkpoints, such as the availability of intermediate checkpoints, by directing them to the dedicated discussion already included in the draft (L197–215). In brief, while we do not claim to resolve every constraint where an unlearning method might be needed, we outline realistic scenarios where our method would be very useful.  We also provide a new strategy for handling duplicate unlearning targets (a consideration not addressed in prior work thus far), and empirically show it can be effective.


We respectfully ask that you consider these points—and the strong consensus among those reviewers who engaged with our clarifications—when making your final decision.



Quick summary of our main contributions:

1. **Practical LLM-scale unlearning via model state arithmetic.**
We propose Model State Arithmetic (MSA), a method that leverages pre-exposure checkpoints to construct effective forget vectors for unlearning in billion-parameter LLMs trained on trillions of tokens. We demonstrate MSA’s effectiveness using the intermediate checkpoints from a standard open-source model, showing that MSA can be applied post-hoc with the checkpoints model providers typically retain.

2. **Advancing data-level unlearning, the core objective.**
A core challenge in machine unlearning is data-level unlearning, making the model resemble the ideal retrained model that never saw the forget set. Across three benchmarks and multiple metrics, MSA consistently matches or improves over prior methods and yields models closer to this ideal.

3. **Shedding light on checkpoint utility.**
We show that MSA can be utilized under  a range of checkpoint choices to compute forget vectors, with even early checkpoints showing high effectiveness.

4. **A stepping stone for checkpoint-based unlearning and data attribution.**
Our results highlight the untapped potential of checkpoints (that are currently routinely stored during LLM development), for improving unlearning and data attribution. We view MSA as a  stepping stone toward a broader line of work on checkpoint-based methods.

---

### Meta-Review · Area_Chair_ngTf · 2026-01-02

**Summary:**

This paper proposed an algorithm called Model State Arithmetic to utilize intermediate checkpoints before seeing the forget set for more effective unlearning. There were various concerns raised by the reviewers: similarity to previous literature, especially some very similar papers like "Unrolling SGD" that completely invalidate the paper's claim of first paper using previous checkpoints to compute gradients for forgetting. The author added related work discussions, acknowledged the similarity, which weaken the novelty but mostly addressed the concern. Other concerns include needing to have access to "clean" checkpoints before seeing the forget set being too restrictive. The authors acknowledged this limitation, but mentioned that in practice, it does not strictly need to be perfectly "clean" checkpoints. There were also concern of lack of theoretical analysis, which the author acknowledged but emphasized that this is mostly an empirical study.

**Reviewer Concerns:**

Most of the concerns are not major and many of them are clarified or at least partially addressed (see summary in the previous box).

**Reviewer Scores:**

I expect the reviewers (except reviewer bmvB, who already gave 8) to slightly raise their scores or keep their current ones as they generally did not have major concerns and the authors provided clarifications that at least partially addressed most of their minor concerns.

---

### Decision · Program_Chairs · 2026-01-26

Accept (Poster)